# Large-scale dynamics of perceptual decision information across human cortex

Niklas Wilming [1✉], Peter R. Murphy [1], Florent Meyniel [2] & Tobias H. Donner [1,3,4,5✉]

Perceptual decisions entail the accumulation of sensory evidence for a particular choice towards an action plan. An influential framework holds that sensory cortical areas encode the instantaneous sensory evidence and downstream, action-related regions accumulate this evidence. The large-scale distribution of this computation across the cerebral cortex has remained largely elusive. Here, we develop a regionally-specific magnetoencephalography decoding approach to exhaustively map the dynamics of stimulus- and choice-specific signals across the human cortical surface during a visual decision. Comparison with the evidence accumulation dynamics inferred from behavior disentangles stimulus-dependent and endogenous components of choice-predictive activity across the visual cortical hierarchy. We find such an endogenous component in early visual cortex (including V1), which is expressed in a low (<20 Hz) frequency band and tracks, with delay, the build-up of choice-predictive activity in (pre-) motor regions. Our results are consistent with choice- and frequency-specific cortical feedback signaling during decision formation.

---

[1] Section Computational Cognitive Neuroscience, Department of Neurophysiology and Pathophysiology, University Medical Center Hamburg-Eppendorf, Martinistrasse 52, Hamburg 20251, Germany. [2] University Paris-Saclay, Inserm, CEA, NeuroSpin, Cognitive Neuroimaging Unit, 91191 Gif-sur-Yvette, France. [3] Bernstein Center for Computational Neuroscience, Charité Universitätsmedizin, Haus 6, Philippstraße 13, 10115 Berlin, Germany. [4] Department of Psychology, University of Amsterdam, Weesperplein 4, 1018 XA Amsterdam, The Netherlands. [5] Amsterdam Brain and Cognition, University of Amsterdam, Nieuwe Achtergracht 129, 1018 WS Amsterdam, The Netherlands. ✉email: niklas.wilming@gmail.com; t.donner@uke.de

A fundamental issue in neuroscience is to understand the mechanisms underlying decisions about the state of the sensory environment. Convergent progress in computational theory, behavioral modeling, and neurophysiological analysis has converged on an influential framework for perceptual decisions: sensory evidence supporting a particular choice is accumulated over time into an internal decision variable; in many contexts, this decision variable is reflected in the motor plan to report the corresponding choice[1–4]. Computational models based on these notions perform well in fitting behavioral choice, reaction times, and associated confidence, of different species in a variety of behavioral task protocols[1,4,5]. Specifically, when the duration of available perceptual evidence is controlled by the environment, the sign of the decision variable at the end of the evidence sequence determines choice, and its magnitude combined with elapsed time[6], determines the associated confidence.

The above class of models affords an intuitive interpretation of neural signals that have been observed at different processing stages of the cerebral cortex of several species. The sensory evidence relevant for a given choice task is encoded by sensory cortical neurons[2]. By contrast, neural correlates of accumulated sensory evidence have been identified in the build-up of motor preparatory activity in different regions of the rat[7], monkey[2,3,6,8], and human cerebral cortex[9–12]. These results are in line with the idea that sensory responses are fed into an integrator on their way to associative and (pre-) motor cortical circuits[2,13].

Yet, the large-scale cortical organization of perceptual decision computations has remained elusive, for a number of reasons. First, only few studies have assessed the dynamics of sensory and choice-related neural signals across multiple cortical areas[14–17]. Second, such large-scale neural dynamics have not been related to the time course of evidence accumulation inferred from behavior—which is, in turn, critical for their computational interpretation[18–20]. Third, the above framework entails a purely feedforward accumulation of sensory evidence across the cortical sensory-motor pathways. In this view, decision-related neural activity in sensory cortex exclusively reflects the feedforward impact of sensory evidence on choice[21,22]. By contrast, the large-scale network implementing the transformation from sensory input to choice is equipped with powerful feedback connections from association and (pre)-motor cortex to sensory cortex[23,24]. In hierarchical circuit models of decision-making, these feedback connections continuously propagate the evolving decision variable from downstream regions to sensory regions[19,20]. As a consequence, the choice-predictive activity observed in sensory cortical areas should be a mixture of a stimulus-dependent (feedforward) and an endogenous (feedback) component[19,20]. Recent single-unit results from monkey visual cortex are consistent with such decision-related feedback[18,25], but the source of this feedback remains unknown.

Identifying the neural implementation of perceptual decision-making requires an integrated behavioral and large-scale physiology approach. Controlled fluctuations in momentary sensory input are instrumental for inferring the time course of evidence accumulation from behavior, and for pinpointing the nature of decision-related neural signals[4,7,19,20]. Such stimulus fluctuations help un-mix stimulus-dependent and putative endogenous components of choice-predictive neural activity[18], within different stages of the sensory cortical hierarchy.

Here, we develop such an integrated approach in the human brain. We combine a visual choice task with behavioral analysis of the underlying evidence accumulation dynamics and an atlas-based, regionally specific magnetoencephalography (MEG) decoding technique in humans. This approach enables tracking the dynamics of (i) sensory evidence weighting on behavioral choice, (ii) the encoding of the instantaneous sensory input and its temporal integral, and (iii) build-up across of choice-predictive activity. This illuminates the large-scale dynamics of perceptual decision information across the mosaic of functional regions spanning the cortical surface. We find an endogenous component of choice-predictive activity in early visual cortex (including area V1), which is expressed in a low (<20 Hz) frequency band and tracks the build-up of choice-predictive activity in (pre-) motor regions, with about 150 ms delay.

## Results

**Temporal profile of evidence weighting for behavioral choice.** Our task required participants ($N = 15$) to compare the mean contrast of a so-called test stimulus with the contrast of a previously presented reference stimulus that was constant across trials (Fig. 1a and "Methods"). The stimuli were circular gratings spanning the entire projection screen (radius >10° of visual angle) and expanded or contracted on a given trial (no changes of direction within trials). The reference contrast was 50% on all trials. The test had a mean contrast that was stronger or weaker than the reference contrast (stimulus category randomly chosen per trial; mean continuously adjusted to 75% accuracy through a staircase procedure). Critically, the test was made up of a stream of ten samples (100 ms each) with contrast levels that fluctuated around the mean contrast for the trial, requiring participants to accumulate the fluctuating contrast values over time in order to compute their mean. The test stimulus offset prompted participants to deliver their behavioral report by button press. They responded with the left or right hand to report "test is stronger than reference" (henceforth called stronger) choices, or "test is weaker than references" (weaker) choices, and with the index or middle finger to report that their confidence about the accuracy of that choice was "high" or "low", respectively.

A number of observations indicate that participants indeed accumulated contrast information across the complete test stimulus interval (i.e., all ten samples, Fig. 1b, c and Supplementary Fig. 1). First, participants' choices were reliably predicted by mean test contrast in all participants (mean prediction accuracy: 76%, range 61–81%; stratified 5-fold cross-validated logistic regression), and their confidence judgments lawfully depended on average test contrast, exhibiting signatures established in other tasks (Supplementary Fig. 1).

Second, contrast information at all sample positions had a significant leverage on behavior (Fig. 1b, c). Contrast fluctuations affected both choice (Fig. 1b; compare curves for stronger vs. weaker choices), as well as confidence reports (high vs. low, for the same choice). We used psychophysical reverse correlation[18,26–28] to quantify the time course of the weighting of contrast information on choice. The evidence weighting time courses, so-called psychophysical kernels, were computed as the area under curve (AUC) of the receiver-operating characteristic relating a given contrast level to the participant's choice (ref. [29]; see "Methods" for details). The AUC ranged between 0 and 1, whereby values of 0.5 indicated chance level. AUC values for all sample positions were significantly >0.5 (Fig. 1c), indicating that sample contrasts stronger than reference tended to be followed by stronger choices (conversely for AUC < 0.5). Critically, however, the impact of contrast samples on choice declined over time across the test stimulus interval (Fig. 1c; slope of psychophysical kernel: $-0.009$, $t(14) = -5.3$, $p = 0.0001$). This is consistent with results from a range of other perceptual choice tasks in humans and monkeys[26–28,30], and provides a reference for interpreting the dynamics of decision-related neural activity below.

**Dynamics of encoding of sensory input across cortex.** We reconstructed the dynamics of sensory and decision-related

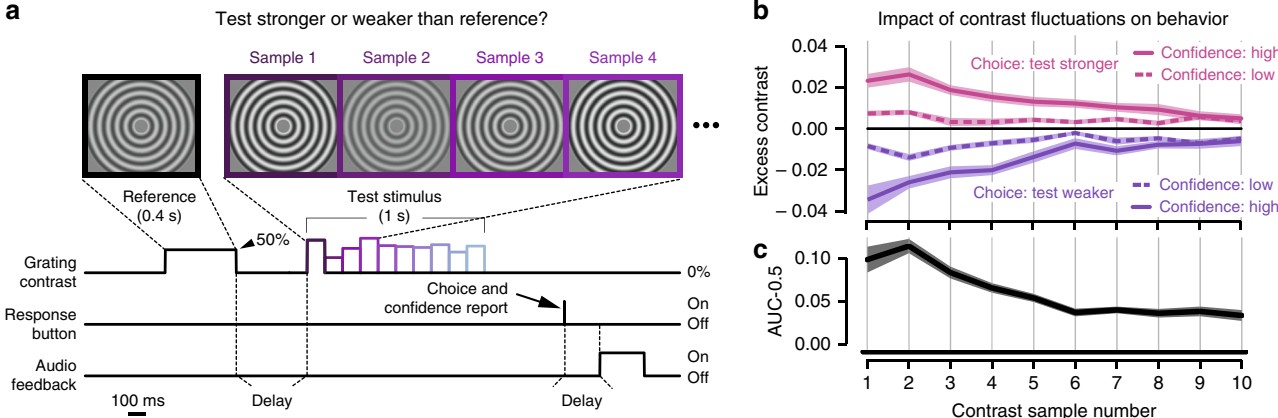

**Fig. 1 Behavioral task and temporal accumulation of sensory input for decision-making. a** Schematic of task events during example trial. Each trial started with a reference stimulus, a circular moving (expanding or contracting) grating with constant contrast (50% in all trials), followed (1–1.5 s delay) by a test stimulus. The test was another circular moving grating made up of ten consecutive samples of fluctuating contrast. Sample contrasts were drawn from a normal distribution ("Methods"). The task was to indicate if the mean test contrast (averaged across samples) was stronger or weaker than the reference contrast. Auditory feedback (low or high tone) was provided after another variable delay (0–1.5 s). Gratings depicted here have lower spatial frequency than in the experiment for visualization purposes. **b** Impact of contrast fluctuations on behavior. Top, contrast fluctuations around the sample mean (i.e., expected contrast on sample s) sorted by behavioral response (choice and confidence judgment). Bottom, psychophysical kernel quantifying impact of trial-by-trial contrast fluctuations on choice, as a function of sample position, expressed as deviation of area under the ROC curve from chance level (AUC = 0.5, see "Methods"). Data are represented as mean ($n = 15$ subjects, lines) and ±SEM (shaded areas). Black horizontal bar, $p < 0.05$ (FDR-corrected two-sided $t$-test of AUC different from 0.5). Source data are provided as a Source Data file.

activity in many cortical regions of interest (ROIs, "Methods"). We focused on a set of functional magnetic resonance imaging (fMRI)-defined regions that are implicated in the visuo-motor decision process for detailed characterization of the properties of the physiological signals reflecting sensory input and ensuing decision (Fig. 2 and "Methods"). This set consisted of multiple topographical maps of the visual field (see "Methods" for area labels) and three regions exhibiting hand movement-specific lateralization: anterior intraparietal sulcus (aIPS), the junction of the IPS and postcentral sulcus (IPS/PostCeS), and the hand-specific sub-region of primary motor cortex (henceforth called M1-hand). In complementary analyses, we tracked sensory and decision-related activity (through decoding of spectral and spatial patterns) in a set of 180 regions per hemisphere covering the entire cortex (Fig. 3 and "Methods").

Previous work has shown that population activity in visual cortex scales monotonically with stimulus contrast in a high-frequency range, including the gamma-band (about 40–70 Hz)[31–33]. Correspondingly, the power in a narrow-band gamma-band (45–65 Hz), as well as a broader high-frequency band (65 to about 120 Hz, high-frequency), was elevated relative to pre-stimulus baseline (Supplementary Fig. 2a) throughout stimulus presentation, in all visual field maps, but not in the movement-selective regions (Fig. 2a and Supplementary Fig. 2b). By contrast, alpha-/beta-band (about 8–36 Hz) activity was suppressed during the test stimulus in all areas, including the movement-selective ones (Fig. 2a and Supplementary Fig. 2b). The power responses in the gamma- and high-frequency-bands, but not in alpha- and beta-bands, also differentiated between trials with stronger and weaker test contrast (Fig. 2b and Supplementary Fig. 2c), thus reflecting the two mean stimulus categories judged by the participants.

Power responses in the gamma- and high-frequency-bands decayed monotonically during test stimulus presentation (Supplementary Fig. 2b, c), with a slope that depended on the variance of sample-to-sample contrast fluctuations (Supplementary Fig. 3; V1: $F(2,28) = 3.92$, $p = 0.03$, one-way ANOVA). This observation is in line with an attenuating effect of contrast adaptation on visual cortical responses observed in previous work[34]. Critically, this decay of visual cortical responses, at least

in part, explained the decay in evidence sensitivity throughout the trial that was evident in behavior: the individual time courses of gamma- and high-frequency-band power were strongly correlated with the time courses of psychophysical kernels (Supplementary Fig. 4).

Responses in the gamma- and high-frequency bands also specifically tracked the rapid sample-to-sample contrast fluctuations throughout the trial (Fig. 3). We trained pattern classifiers to decode sample contrast from local spectral patterns (power values from 1 to 145 Hz) and correlated decoded with physical sample contrasts ("Methods"). Contrast decoding profiles in V1 for consecutive samples peaked about 190 ms after sample onset (Fig. 3a). The convex hull curve of peak decoding values for individual contrast samples, a summary measure of decoding performance across samples, was significantly better than chance throughout decision formation (Fig. 3b, orange). Just like the gamma-band and high-frequency responses, the precision of V1 contrast decoding also decreased monotonically from the first to the ninth sample (Fig. 3b left), tracking the decay of overall gamma-band and high-frequency responses (Supplementary Fig. 5). A complementary encoding analysis (using frequency-resolved regressions assuming predominantly linear encoding of contrast in MEG power, see "Methods" and "Discussion"), confirmed that the gamma-band contributed most to the contrast decoding in V1: regression weights were strongest in the gamma-band and significant only for the gamma- and high-frequency-bands (Fig. 3c). In sum, neural population responses in V1 and extrastriate visual cortical areas conveyed detailed information about the sample-to-sample contrast fluctuations, which was specifically contained in the gamma- and high-frequency-bands.

**Dynamics of encoding of decision variable and choice across cortex.** When choices are reported with left- or right-hand movements (as in our task), the hemispheric lateralization of activity in motor and parietal cortical regions encodes specific choices[9,35,36]. This activity is frequency-specific and builds up during decision formation[9,10], evident in the power lateralization

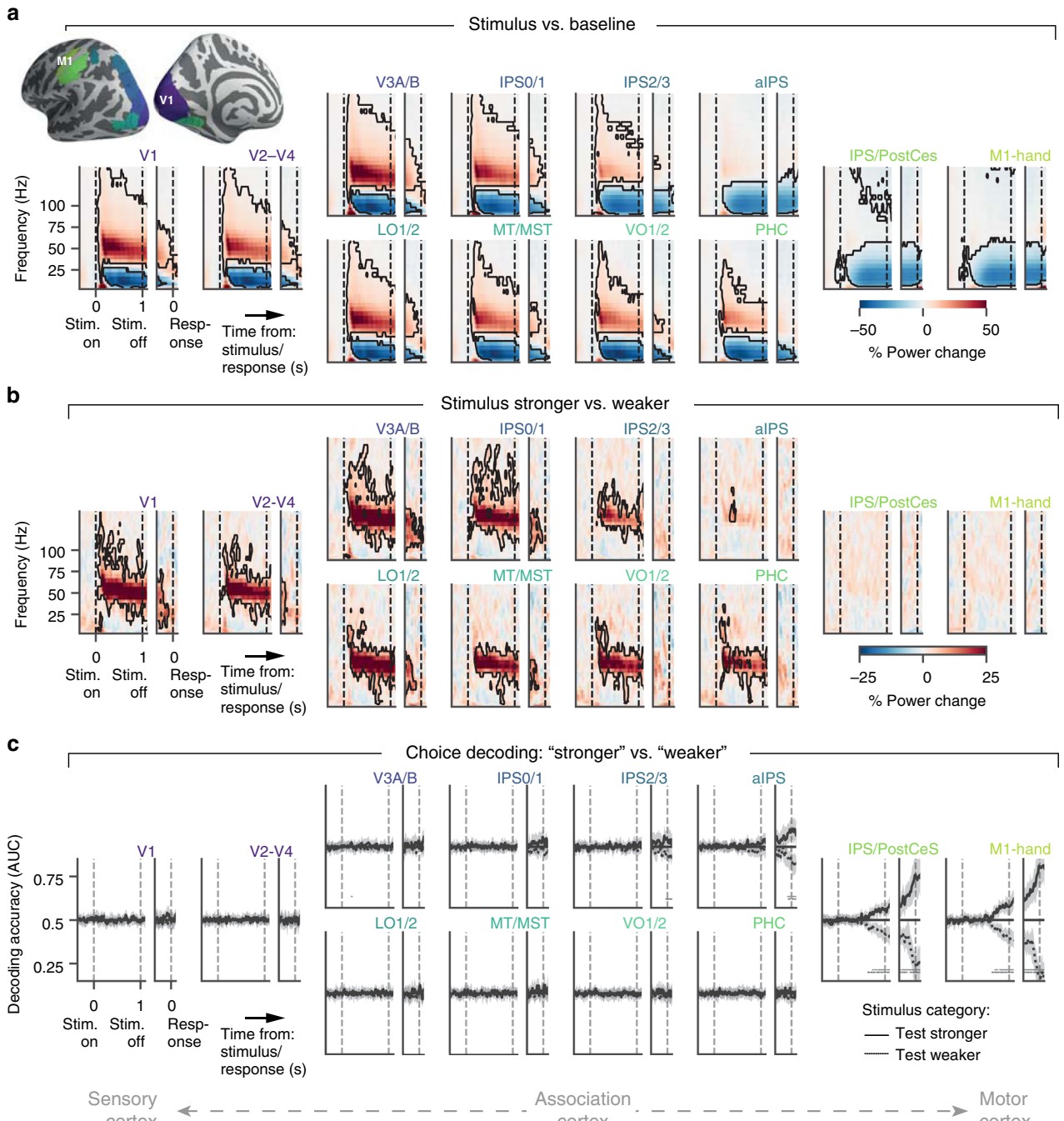

**Fig. 2 Task-related neural dynamics across the visuo-motor cortical pathway. a** Time-frequency representations (TFRs) of group average power change relative to pre-stimulus baseline (−250 to 0 ms relative to test stimulus), for several cortical regions along the cortical sensory-motor axis. Top left inset, colors of areas depicted on cortical surface. For each area, data are shown time-locked to test stimulus-onset in the left partition of each TFR and time-locked to the choice, in the right partition (partitions separated by white spacing). Data are represented as mean (n = 15, color code). Black contour, p < 0.05 (cluster-based, two-sided permutation test against 0). **b** As **a**, but for power differences between stronger and weaker test stimulus categories. **c** Time courses of regionally specific decoding of choices (see "Methods"), separately for the two test stimulus categories. Data are represented as mean (n = 15 subjects, lines) and 95% highest density interval (shaded areas) of posterior distributions over group average decoding accuracies. Horizontal bars, p < 0.05 (cluster-based permutation test: AUC different from 0.5, n = 15 subjects). Source data are provided as a Source Data file.

(contra- vs. ipsilateral to effector choice) of M1-hand in our data (Supplementary Fig. 6a). Correspondingly, decoders trained on the spectral power profiles from both hemispheres revealed choice-predictive build-up activity in M1-hand, as well as IPS/PostCeS (Fig. 2c). Choice decoding in these regions ramped up more quickly before choices associated with high than with low confidence (Supplementary Fig. 6b).

Choice-predictive activity in IPS/PostCeS was partially independent of M1-hand: When eliminating signal leakage by removing M1-hand power from IPS/PostCeS power for each frequency (via linear regression), the residual IPS/PostCeS activity still yielded significant choice decoding (AUC = 0.53, time = 1.1 s; t(14) = 2.01, p = 0.032). Movement-specific activity in IPS/PostCeS has previously been observed with fMRI during

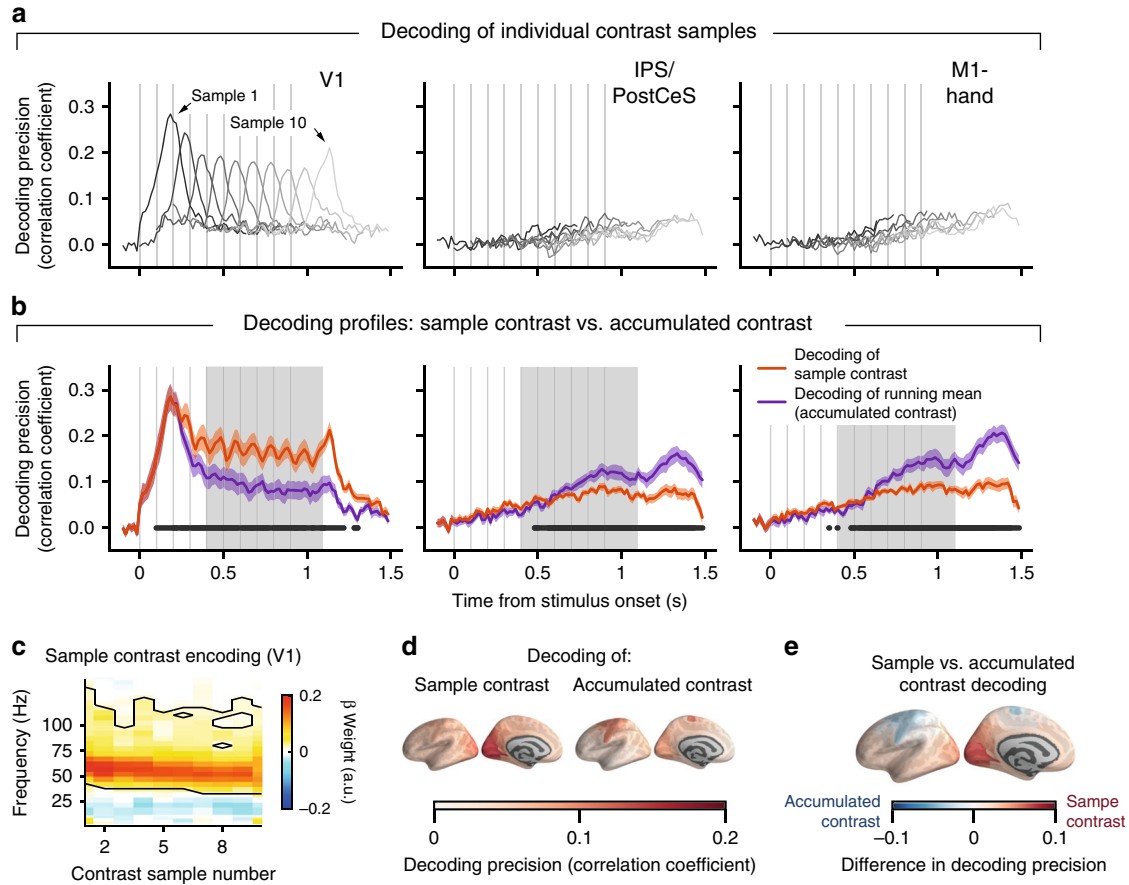

**Fig. 3 Decoding of sample contrast and accumulated contrast across cortex. a** Time courses of frequency-based (range: 1–145 Hz; V1: averaged across hemispheres, IPS/PostCeS/M1-hand: lateralized power values) decoding of single-trial contrast values, separately for each sample position, from V1 (left), IPS/PostCeS (middle) and M1-hand (right) power responses. Decoding precision is expressed as cross-validated Pearson correlation coefficient between decoded and presented contrast (see "Methods"). Data are represented as mean ($n = 15$ subjects, lines). **b** Decoding of sample contrast and of running mean of contrast. Orange lines, convex hull across decoding peaks for individual samples in panel **a**. Magenta lines, convex hull across decoding time courses for the running means of sample contrast values up to sample $i\epsilon\{1,...,10\}$. This reflects the neural representation of the decision-relevant quantity in the task. Data are represented as mean ($n = 15$ subjects, lines) and ±SEM (shaded areas). Horizonal bar: $p < 0.05$ (cluster-based permutation test of decoding of sample contrast vs. decoding of accumulated contrast). **c** Encoding of individual contrast samples as a function of frequency. Data are represented as mean beta weights ($n = 15$ subjects, color code). Black contour, $p < 0.05$ (cluster-based, two-sided permutation test). **d** Group average decoding precision ($n = 15$ subjects, convex hull averaged across gray shaded areas in B) for all ROIs: single-contrast samples (left), accumulated contrast (center), and their difference (right). Source data are provided as a Source Data file.

perceptual choice[36], but the build-up of activity in this region during evidence accumulation, prior to overt behavioral choice, has not yet been established.

The above analyses focused on a set of cortical regions, in which the format of choice-predictive activity in terms of motor preparatory activity is well understood (see above and Supplementary Fig. 2). Two complementary analyses applied the same choice decoding approach as for Fig. 2c to 180 regions spanning the entire cortical sheet (Supplementary Fig. 7a), as well as an alternative choice decoding approach (using more fine-grained spatial information as well as signal phase and amplitude, "Methods") to a set of pre-selected frontal regions based on previous work (Supplementary Fig. 7b). This showed the strongest choice-predictive activity in motor cortex (M1 and PMd) supporting our focus on action-related regions (see Discussion).

Activity in motor and parietal cortex (M1 and IPS/PostCeS, respectively) tracked not only the evolving plan to act (choice-predictive activity), but also the ongoing computation of the decision-relevant quantity: the mean of contrast samples. We trained pattern classifiers to decode the running mean of contrast

samples at a specific latency after sample onset (dubbed accumulated contrast in Fig. 3; see "Methods"). The correlation between the decoded and actual accumulated (running mean of) contrast quantified the precision of accumulated contrast read-out from activity patterns (magenta lines in Fig. 3b). Decoding values late in the trial (after gray box) were likely affected by the hand movement execution, but movement execution could not explain decoding before 1.1 s after stimulus onset: Our analyses excluded reaction times faster than 1.225 s and used a maximum time window of 250 ms for spectral estimation; thus, 1.1 s corresponded to the fastest reaction time minus half the spectral estimation time window. Also note that the observed decoding of accumulated contrast should be considered a lower bound on the expression of the hypothetical decision variable in cortex: the analysis assumed perfect accumulation of all contrast samples across time, so any deviation from this assumption (e.g., due to reduction in evidence sensitivity, Fig. 1b) reduced decoding precision.

Mapping the sensitivity of neural responses to instantaneous or accumulated contrast (average of convex hulls across the interval 0.4 to 1.1 s, gray box in Fig. 3b) across the cortical surface also

showed strongest decoding of individual contrast samples for early and intermediate visual cortical areas (with peak reliability in V1), whereas decoding of accumulated contrast prevailed in the same action-related regions of parietal and (pre-)motor cortex (Fig. 3d, e), in which build-up activity also predicted behavioral choice (compare Supplementary Fig. 7a). Overall, the functional properties of IPS/PostCeS, PMd, and M1 activity resembled those of monkey and rodent parietal and frontal cortical regions during similar perceptual choice tasks[2,7,8]. Mapping the dynamics of stimulus- and choice-encoding across the cortical surface yielded results in line with the idea that the action plan encoded in these parietal and frontal cortical regions was computed by accumulating the contrast samples over time.

**Untangling decision-related activity in visual cortex.** We next sought to investigate the dynamics of choice-predictive activity in more detail for visual cortical areas. The temporal profile of choice-predictive activity in sensory cortex, combined with the temporal profile of sensory evidence weighting (i.e., psychophysical kernel), can help reverse-engineer decision-related interactions between processing stages involved in evidence encoding and evidence accumulation, respectively[19,20]. This choice-predictive activity refers to the trial-to-trial correlation between neural-activity fluctuations within a stimulus category and behavioral choice. It is commonly dubbed as choice probability for single-unit activity, yielding small but significant values for early visual cortex[18,21,22,37]. We quantified this activity in the same way as psychophysical kernels, but replacing sample contrast with the band-limited power in different visual cortical areas ("Methods"). While this analysis is analogous to the one used to calculate single-unit choice probabilities, we refer to the resulting measure as V1 kernels to highlight the difference to single-unit spiking activity (see "Discussion").

Visual cortical gamma-band responses encoded the fine-grained contrast information (Figs. 2b and 3) that subjects used for solving the task (Fig. 1 and Supplementary Fig. 1). We thus reasoned that choice-predictive fluctuations of gamma-band activity should mirror the impact of visual contrast information on choice inferred from behavior (Fig. 1C). Indeed, V1 gamma-band kernels (40–75 Hz, range based on Fig. 2b) were larger than zero (i.e., AUC > 0.5) early in the trial (Fig. 4a, left, overall kernels in blue, $p < 0.05$ for samples 1 and 2 only), but neither during the baseline interval before the test stimulus (Fig. 4b, left; $t(14) = -1.24$, $p = 0.24$), nor late in the trial (Fig. 4a). V1 gamma-band

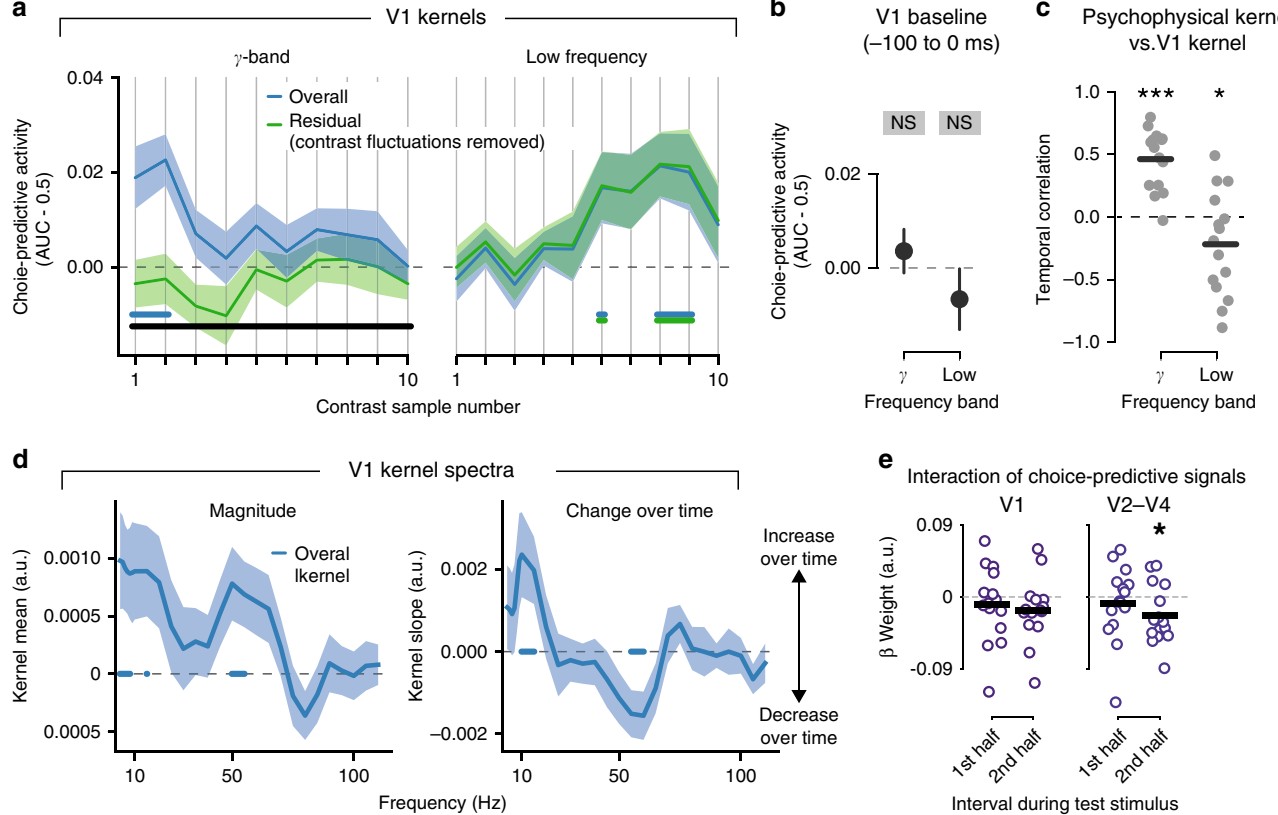

**Fig. 4 Dissociating sensory and endogenous components of choice-predictive V1 activity. a** Kernels quantifying the impact of hemisphere-averaged V1 power fluctuations (within each test stimulus category) on behavioral choice (V1 kernels) for V1 activity in the gamma-band (40–75 Hz, left) and low-frequency (0–20 Hz right). Power values have been extracted at the latency of peak contrast encoding ($t = 190$ ms from sample onset). Blue, overall kernels, based on all fluctuations in V1 power. Green, residual kernels, based on residual fluctuations after removing effect of external stimulus fluctuations (see main text). **b** Test for choice-predictive gamma-band and low-frequency activity during pre-stimulus baseline interval. **c** Correlation of V1 kernels with psychophysical kernel from Fig. 1B. Each dot is one participant. Black lines, group average. $\gamma$, $t(14) = 6.5$, $p = 1.3 \times 10^{-5}$; low, $t(14) = -2.19$, $p = 0.045$, $n = 15$ subjects. **d** Frequency spectra of choice-predictive activity in V1. Left, kernels collapsed across time. Right, change of kernels across test stimulus interval (slope across samples). **e** Interaction of impact of low-frequency and gamma-band activity on behavioral choice, for areas V1 and V2–V4. V2–V4, 2nd half: $t(14) = -2.4$, $p = 0.0305$. Data in **a**, **d** are represented as mean ($n = 15$ subjects, lines) and ±SEM (shaded areas); in **b** are mean ($n = 15$ subjects, dots) and ±SEM (error bars); in **c**, **e**, as mean ($n = 15$ subjects, horizontal lines) and individual subjects (dots). All $p$-values from two-sided $t$-tests. Horizontal bars in **a**, **d**: $p < 0.05$ (colored bars, difference from 0; black bars, difference between blue and green). Asterisk *$p < 0.05$; ***$p < 0.001$; Source data are provided as a Source Data file.

kernels decayed over time (group average slope: $-0.017$, $t(14) = -3.1$, $p = 0.007$) just like the psychophysical kernels, yielding a highly significant temporal correlation (Fig. 4c, left; group average correlation: $r = 0.45$).

In circuit models with feedback from the accumulation stage, choice probabilities in sensory cortex reflect a mixture of feedforward and feedback components[19,20]. These components can be disentangled by removing the contribution of stimulus fluctuations to neural activity[18,19]. If choice-predictive activity fluctuations reflect the impact of external stimulus fluctuations on choice, removing the stimulus-induced fluctuations should reduce (or even cancel) the correlation between neural activity and choice[19]. This is exactly what we found for V1 gamma-band activity: The residual kernels (computed after removing effect of contrast fluctuations via linear regression, "Methods") did not differ from zero (green in Fig. 4a, left; $p > 0.05$ for all sample positions) and overall kernels (blue) were consistently larger than the residual kernels (green) for all sample positions (Fig. 4a, left, black bar).

V1 choice-predictive activity in the low-frequency range (0–20 Hz) was markedly different from the choice-predictive activity in V1 gamma-band (Fig. 4a, right, blue). Trial-to-trial fluctuations in V1 low-frequency power also predicted choice variability, but only for later samples (Fig. 4a), and not during the pre-stimulus baseline (Fig. 4b; right; $t(14) = -1.32$, $p = 0.21$). Accordingly, the time courses of V1 low-frequency kernels and psychophysical kernels were *negatively* correlated (Fig. 4c, right; group average correlation: $r = -0.23$; difference to gamma-band: $t(14) = 5.88$, $p = 0.00005$). What is more, removing contrast-driven fluctuations from low-frequency power did not change the magnitude and shape of V1 low-frequency kernels (Fig. 4a, right, green). Thus, the choice-predictive fluctuations in low-frequency power resulted from endogenous sources, rather than from the external stimulus. Similar results were found across the visual cortical hierarchy (Supplementary Fig. 8). In sum, choice-predictive fluctuations of low-frequency and gamma-band activity dissociated in terms of both their temporal profiles (primacy vs. recency) and sources (external stimulus vs. internal).

To delineate the spectral profiles of the above-described effects, we also analyzed choice-predictive activity across a broad frequency range as a function of frequency. Indeed, overall choice-predictive activity (collapsed across samples) was confined to the low-frequency and gamma-bands (Fig. 4d, left), and the kernel slope (i.e., change over time) showed an *increase* in V1 kernel magnitude over sample positions, specifically for the alpha-band (around 10 Hz, center frequency for computation of low-frequency kernels) and a *decrease* in the gamma-band (Fig. 4d, right). Thus, the functional dissociation between different components of decision-related V1-activity was expressed in clearly delineated frequency bands.

While having dissociated functional properties, endogenous and stimulus-dependent choice-predictive activity components, in low-frequency and gamma-bands, respectively, might interact, somewhere in the visual cortical hierarchy. To test this possibility, we jointly regressed power in these two bands on behavioral choice (logistic regression, "Methods"). We quantified the decaying vs. increasing temporal profiles for both bands, as the asymmetry of beta weights between first and second halves of the test stimulus interval. In line with the opposite kernel slopes in our more fine-grained analysis from Fig. 4d, we found opposite temporal profiles of choice-prediction regression coefficients in both bands (Supplementary Fig. 9). Critically, an interaction term included in the regression was negative for the second half of the decision interval in V2–V4 (Fig. 4e). There was a similar trend in V1: $t(14) = -1.6$, $p = 0.127$. This suggests that low-frequency activity suppressed the impact of early visual cortical

gamma-band responses on choice towards the end of decision formation, when the choice-predictive low-frequency activity in visual cortex was particularly strong.

**Coupled dynamics of decision-related neural activity.** The choice-predictive activity evident in V1 low-frequency power might reflect recurrent dynamics within visual cortex and/or feedback of decision-related signals from downstream regions outside of the visual system[19,20]. The build-up of the group average time course of V1 low-frequency kernels (Fig. 4a, right) resembled the build-up of choice-predictive activity in downstream regions (Fig. 2c), consistent with the feedback scenario. To quantify this relationship, at the level of individual subjects, we re-computed visual cortical low-frequency kernels at the same temporal resolution as downstream choice decoding signals (Supplementary Fig. 10) and correlated the time courses, for a range of different time lags (see Fig. 5, top left inset and "Methods"). For all early visual field maps (V1 and V2–V4), the group average cross-correlations were robust for M1 leading (but not following) visual cortex (Fig. 5, black bars) with a peak lag of 150 ms.

The positive, non-zero peak lags are too large to be explained by signal processing confounds (Supplementary Fig. 11 and Supplementary Discussion), and they argue against signal leakage as the source of the correlation. As a reference for leakage, we quantified the zero-lag correlation between choice-predictive activity and low-frequency kernel, now both taken from M1 (Fig. 5, green data point on top of M1-V1). This so-called reference correlation constituted an upper bound of the leakage that might have been present in the estimates of areas other than M1, thus overestimating the true leakage component present in the visual field map data. Despite the conservative nature of this reference, the cross-correlations with visual cortex at positive lags were significantly larger for several field maps including V2–V4 (Fig. 5, colored bars). These results indicate that the cross-correlations for visual field maps reflect genuine coupling between choice-predictive neural signals in motor and visual cortex.

The positive, non-zero peak lags observed here are consistent with feedback but do not necessarily reflect direct (monosynaptic) interactions between choice-predictive activity in (pre-)motor regions and the visual cortical areas (other areas might have relayed the signals). Our results also do not rule out a contribution of interactions within visual cortex to the choice-predictive low-frequency fluctuations in V1 and other visual cortical areas. Even so, our results show that decision-related visual cortical activity in the low-frequency closely tracked the build-up of decision-related activity in downstream brain regions involved in action preparation.

## Discussion
The neurobiology of perceptual decision-making has witnessed considerable advances in the integration between computational theory and physiological and psychophysical experimentation[1,2]. The field's dominant framework entails the sustained accumulation of transient sensory signals encoded in sensory cortex on the way to association and motor cortex, which is transformed into an action plan (but see refs. [16,38,39]). This framework has lacked a detailed characterization of the above-described computation across the large mosaic of interconnected regions of the primate cortex and ignored the role of abundant feedback connections from association and (pre-)motor cortical regions to sensory cortex.

Here, we developed an approach for tracking the dynamics of (instantaneous and accumulated) stimulus- and choice-related

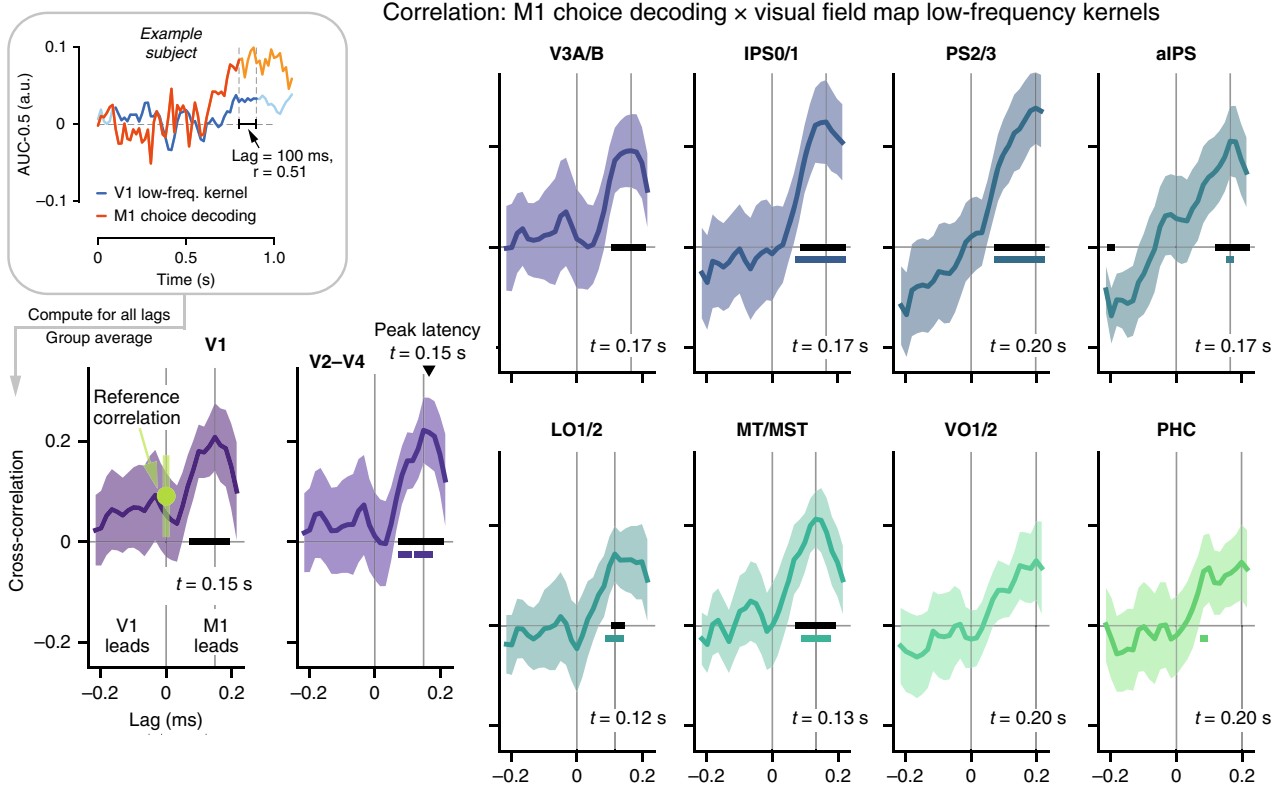

**Fig. 5 Link between choice-predictive M1-activity and visual cortical low-frequency kernels.** Cross-correlation between choice-predictive activity in M1-hand (Fig. 2c) and V1 low-frequency kernel, with the latter re-computed at the same temporal resolution as the former (Supplementary Fig. 10 and inset). Inset: Illustration of analysis approach for one example subject and lag. The cross-correlation function between V1 low-frequency kernel and choice-predictive M1 activity was evaluated for a range of lags from −215 to 215 ms (positive lags indicate lead of M1), and collapsed across subjects. Reference correlation is correlation between choice-predictive activity and low-frequency kernel at lag $t = 0$, both taken from M1 (upper bound of signal leakage); it is shown in V1 plot (green) but was applied to evaluate correlations for all visual field maps. Data are represented as mean ($n = 15$ subjects, lines, reference correlation: green dot) and ±SEM (shaded areas, reference correlation: error bars). Horizontal bars, $p < 0.05$: black bars, against 0; colored bars, against reference correlation (all two-sided $t$-tests). Source data are provided as a Source Data file.

information across the human cerebral cortex. In a task that entailed the accumulation of visual contrast information over time, we found a predominant involvement of early visual cortex in evidence encoding and a predominant involvement of parietal, as well as (pre-) motor regions in evidence accumulation and action planning. We also found a spectral multiplexing of stimulus-dependent and endogenous components of choice-predictive activity fluctuations in visual cortical population activity: The stimulus-dependent component was expressed in the gamma-band, and the endogenous component in low (<20 Hz) frequencies, peaking in the alpha-band. The latter tracked the build-up of choice-information in the downstream regions involved in action planning.

Our current insights hinged on the ability to track the time course of the encoding of sensory evidence, accumulated evidence, and action plan, from each of a large set of well-delineated cortical regions. To this end, we combined atlas-based MEG source reconstruction with a multivariate pattern classification approach that was based on the spectro-spatial patterns of local activity within each region. Previous electrocorticography work on brain-computer interfaces has highlighted such spectro-spatial patterns as useful features for pattern classification[40]. Our current approach differs from previous decoding approaches for human fMRI and MEG data (Supplementary Discussion) and provides the opportunity to track large-scale information dynamics across cortical areas, analogous to recent animal work[14,15,17,41]. In contrast to these large-scale physiology studies in animals, however, we here quantified the associations between regionally

specific information dynamics and the psychophysically inferred time course of evidence accumulation. This, in turn, constrained their functional interpretation.

Several analyses suggest that the choice-related build-up activity in our task was primarily expressed in the format of motor preparatory activity, rather than in a more abstract format (Supplementary Discussion). This is in line with a large body of work in animals[1,2,7,8] and with the fact that the decision variable could, by design, be mapped directly onto an action plan in our task. Even so, it is possible that M1 hand area and IPS/PostCeS only reflected the outcome of an evidence accumulation process that took place elsewhere (e.g., the striatum[13]), which we failed to detect with our approach.

Our decoding approach was agnostic about the nature and specifics of the neural code used for solving the task[42]. We assume that the decision process reads out the spiking output of visual cortical activity, rather than the amplitude of local field potential fluctuations, including oscillations in the gamma-band[32]. V1 spiking activity also scales with contrast in a mono-tonic fashion, and is generally closely coupled to local field potentials (and the resulting MEG activity) in the gamma-band[33]. However, both signals can dissociate[43]. For example, power and peak frequency scale approximately linearly with contrast across the full range[31,32], whereas V1 spiking saturates at high contrasts[44]. By exploiting the full spectral (and spatial) pattern of power changes, both peak shifts and amplitude changes could be used for visual contrast decoding, rendering our approach generic and robust.

Limitations of our analyses, as well as possible, genuine dissociations between the decoded signal and the decision-relevant neural code, may have caused an underestimation of the true link between visual cortical activity and behavioral choice. Specifically, endogenous (stimulus-independent) fluctuations in the neural stimulus-representation that is read out in the decision should impact behavioral choice[42]. However, we did not detect such a choice-predictive effect in the residual V1 gamma-band activity. Our analysis attempted to isolate these endogenous fluctuations by removing stimulus-related trial-to-trial variations in gamma-band activity via linear regression. This was based on the assumptions that (i) MEG power scaled approximately linearly with contrast and (ii) stimulus-related and endogenous components superimpose linearly. Both assumptions are simplified: contrast response functions in MEG power may not be perfectly linear in all individuals (see ref. [45] for demonstration for motion coherence response functions), and stimulus- and choice-related cortical signals may interact multiplicatively[46]. Consequently, the absence of a choice-predictive component in residual V1 gamma-band activity observed in our analysis may reflect technical limitations, such as a failure of our linear approach to isolate the endogenous fluctuations and/or the low signal-to-noise ratio of MEG gamma-band activity[33]. Alternatively, the absence of this effect may be due to gamma-band activity being only an indirect proxy of the neural code used for the decision computation.

Even so, our results highlight the utility of decomposing the local field potential into different frequency bands for disentangling distinct components of cortical computation. The results also add to the mounting evidence for the co-existence of stimulus-related and endogenous components in the choice-predictive activity of visual cortex[18,19], and they identify physiological markers, which future studies could also assess in the frequency-specific local field potential activity from invasive animal data as well from hierarchical cortical circuit models. While spectral analysis helped to disentangle the stimulus-related and endogenous component of activity, we do not conceptualize these two frequency-specific signals as independent entities. There is abundant evidence for interactions between power in different frequency bands, including alpha and gamma, in the cortex[33,47]. For the choice-predictive power components assessed here, we found a negative multiplicative interaction between the impact of low-frequency and gamma-band activity on choice in areas V2–V4 (Fig. 4f). These choice-specific cross-frequency interactions should be further characterized by means of causal interventions.

Our findings add to the realization that adaptation phenomena in sensory cortex likely contribute to decision-making[4,16]. Attenuation of sensory responses is a well-established phenomenon in various behavioral states, which is often explained by local circuit mechanisms[34,48,49]. Yet, most models of perceptual choice have ignored adaptation. In our task, we found a progressive attenuation of visual cortical gamma-band responses, dependent on sample variance. This progressive attenuation predicted the individual reduction in the impact of evidence on choice evident in psychophysical kernels. Such primacy in psychophysical kernels has often been explained by bounded evidence accumulation at the decision stage[26,28]. In line with ref. [16] (their Fig. 2e), our results indicate that the attenuation of neural responses at sensory processing stages can account, at least in part, for a decrease in evidence sensitivity over time. Note that both mechanisms, bounded accumulation and adaptation, are not mutually exclusive, but might, in fact, interact in the presence of decision feedback.

The observation that stimulus-dependent and endogenous components of decision-related activity in visual cortex were expressed in the gamma-band vs. the low-frequency (alpha) band,

respectively, resonates with an emerging view of the role of the gamma- and alpha-bands in message passing across the cortical hierarchy. In this view, feedforward and feedback information flow through the cortical hierarchy is mediated by layer-specific pathways communicating in these two spectral channels, respectively[24,50–55]. Those studies have quantified the properties of physiological local field potentials recorded from different areas (e.g., time-lagged correlations between regions or laminar activity profiles within regions), without a link to specific behaviors. The functions of these cortical feedforward and feedback pathways have been conceptualized in the context of attention[56] and predictive coding[52,57], but not the accumulation of evidence towards a choice. Evidence accumulation, the essence of dynamic belief updating, may be a useful framework for advancing and testing theories of spectral channels for inference. Our current insights suggest that the spectral profiles of the associated local field potential activity (our MEG source estimates) help disentangle the feedforward and feedback components of decision-related signals in the cortical hierarchy. Future work should assess if, and to what extent, the contribution of gamma-band activity to the feedforward signaling of decision-relevant sensory evidences generalizes to stimuli beyond contrast gratings, which induce strong, narrow-band gamma oscillations (see above). Recent results from a visual evidence accumulation task entailing small checkerboard patches flashed at different positions suggest they do[58].

## Methods

**Participants**. Fifteen participants (8 female, 7 male) participated in the experiment. All had normal or corrected to normal vision and no history or indications of psychological or neurological disorders. The experiment was approved by the ethics committee of the Medical Association Hamburg. Participants gave written and informed consent. Participants were paid 10 € per hour of participation. The sample size and trial number per participant (see next section) was chosen based on previous work with similar designs and neurophysiological[9,45] and psychophysical[27] measures of decision-making.

**Task and procedure**. Each trial of the task consisted of the following sequence of events (Fig. 1a). First, a reference stimulus (grating of fixed contrast at 0.5) was displayed for 400 ms. After a variable delay (uniform between 1 and 1.5 s) ten successive samples of variable contrasts (see below) were shown (each 100 ms); together these ten samples made up the test stimulus, the mean contrast of which participants should compare with the reference (i.e., forced choice report: "test is stronger than reference" or "test is weaker than reference").

The offset of the last sample marked the beginning of the response period for participants. Participants reported their binary choice, and their confidence about the correctness of that choice (high vs. low) simultaneously, by pressing one of four different buttons, whereby the two hands were always mapped to different choices (choice-to-hand mapping counterbalanced across participants). The index and ring fingers of each hand were then used to report confidence (again mapping counterbalanced across participants). During MEG sessions, participants used two response pads, one for each hand. During the training sessions participants, used the same stimulus-response mapping, but pressed keys on a computer keyboard. After a participant's response and a consecutive variable delay between 0 and 1.5 s auditory feedback was given (250 ms duration). A low tone indicated a wrong answer and a high tone indicated a correct answer.

The ten consecutive contrast samples were draws from a normal distribution centered on a participants 75% accuracy contrast level. This threshold was determined by running a QUEST staircase[59] continuously in the background. The standard deviation of the normal distribution was chosen randomly from [0.05, 0.1, 0.15] from trial to trial. After each set of 100 trials participants could take a short break self-timed break. After the third block, participants took a longer break lasting at least five minutes.

Each participant completed five sessions, whereby each session consisted out of five blocks à 500 trials, lasting approximately 60 min. The first session was a training session that took place in a behavioral laboratory and was used to expose participants to the task and to calibrate their performance to 75% correct. The subsequent four sessions were experimental recording sessions that took place in the MEG laboratory and yielded the data analyzed in this paper. Experimental sessions were spread out over several days and were typically completed within 10 days. We also collected a structural MRI for each participant in a separate session, unless an MRI scan was available from previous experiments.

Based on previous work[9,27,45], we estimated that four main experimental sessions per participant would be necessary for obtaining robust

neurophysiological measures of decision-making. Thus, data were collapsed across all experimental sessions, and no replication was attempted in this study.

**Stimuli**. We used expanding or contracting circular gratings similar to the stimuli from Michalareas[51]. The intensity $a$ of a given pixel $(x,y)$ was given by computing a blending value for each pixel:

$$a_{(x,y)} = \tfrac{1}{2} + \tfrac{1}{2}\sin\left(\frac{d_{(x,y)}-s}{2r\pi}\right),\qquad(1)$$

where $d_{(x,y)}$ was the distance of pixel $(x, y)$ to the center of the screen and $r = 3/4°$ determined the spatial frequency of the grating. Varying $s$ over frames yielded expanding or contracting gratings, respectively. We varied $s$ such that the grating moved with a speed of $4/3° \text{ s}^{-1}$. On each trial, the grating either expanded or contracted, but never changed direction. To obtain a final color value we used $a_{(x,y)}$ to blend two grayscale colors that had the desired contrast:

$$l_{(x,y)} = a_{(x,y)}\left(0.5 - \tfrac{c}{2}\right) + \left(1 - a_{(x,y)}\right)\left(0.5 + \tfrac{c}{2}\right),\qquad(2)$$

where $c$ was the desired contrast. We furthermore set an inner annulus with a radius of 1.5° to uniform gray. Gratings had a generative radius of 12.5°, but were truncated by the vertical screen border at a radius of 11.3°. The contrast of the reference grating was always set to 0.5, while the contrast of the test stimulus varied and changed every 100 ms, as controlled by the staircase procedure (see above).

Stimuli were generated using Psychtoolbox 3 for Matlab. They were back-projected on a transparent screen using a Sanyo PCL-XP51 projector with a resolution of $1920 \times 1080$ at 60 Hz. The luminance profile was linearized by measuring and correcting for the systems gamma curve. A doubling of contrast values, therefore, also produced a doubling of luminance differences. During the first training session stimuli were presented on a VIEWPixx monitor with the same resolution and refresh rate (also linearized).

**Data acquisition**. We used a CTF MEG system with 275 axial gradiometer sensors and recorded at 1200 Hz. Recordings took place in a dimly lit magnetically shielded room. We concurrently collected eye-position data with a SR-Research EyeLink 1000 eye-tracker (1000 Hz). We continuously monitored head position by using three fiducial coils. After seating the participant in the MEG chair, we created and stored a template head position. At the beginning of each following session and after each block we guided participants back into this template position. We used Ag/AgCl electrodes to measure ECG and vertical and horizontal EOG.

**Evaluation of choice and confidence dependence on contrast**. We used a logistic regression to evaluate whether participants exploited different contrast samples for their choices and confidence judgments. We fit the following logistic regression to predict choices from contrast values:

$$y_{\text{trl}} = \text{logistic}\left[\sum_{i \in \{1..10\}} \beta_i c_{\text{trl},i} + \beta_0\right],\qquad(3)$$

where $y_{\text{trl}}$ was the choice in trial trl and $c_{\text{trl},i}$ was the contrast value of sample $i$ in the same trial. We evaluated the accuracy of this fit with fivefold cross-validation. All available trials from one subject were split into five folds and we used each fold as test set once and all remaining folds for weight estimation. We carried out cross-validation per subject and then averaged across folds and subjects. Since we titrated participants' accuracy to 75% correct, we also expected that the accuracy of this logistic regression is bounded close to 75% correct. We also evaluated whether confidence judgments were based on contrast. To this end, we fit a similar logistic regression, but this time predicted confidence judgments for each response separately. We again evaluated the accuracy of this logistic regression with fivefold cross-validation.

**Trial categories for the current task**. The computation of psychophysical and neural-activity kernels described below required sorting trials into four categories defined by a unique combination of the physical stimulus category (i.e., mean contrast of test stimulus stronger or weaker than reference) and the participant's perceptual choice (stronger or weaker). We defined these four categories based on signal detection-theory[29], as follows. Hits and misses: stronger and weaker choices, respectively, for trials in which the physical test stimulus was stronger than the reference; false alarms and correct rejects: stronger and weaker choices, respectively, for trials in which the physical test stimulus was weaker than the reference.

**Computation of psychophysical kernels**. With the term psychophysical kernel, we refer to the time course of the correlation between trial-to-trial fluctuations in stimulus sample contrast and the participant's behavioral choice, after factoring out the physical stimulus category (i.e., mean test contrast stronger or weaker than reference). Psychophysical kernels were computed by comparing single-trial sample contrast values at a given sample position, between both behavioral choices, within a given stimulus category (test contrast stronger or weaker than reference). To this end, we computed the receiver-operator-characteristic curve (ROC), separately for each stimulus category and sample position, and from this the area under the ROC curve. The resulting AUC values (range: 0–1) quantified the

predictive power of contrast fluctuations around their mean level within a stimulus category for choice. AUC values of 0.5 indicated that contrast fluctuations did not differ between choices. AUC values >0.5 indicated that larger contrast values tended to be followed by stronger choices (with AUC = 1 indicating perfect separation of contrast values between choices), and AUC values <0.5 indicated that larger contrast values tended to be followed by weaker choices. We computed separate AUC time courses by comparing (i) hits and misses and (ii) false alarms and correct rejects. The resulting two AUC time courses were averaged, yielding the single psychophysical kernel shown in Fig. 1c.

**Analysis and source reconstruction of MEG data**. We used beamforming to reconstruct the sources of activity observed at the MEG sensor level. First, we automatically labeled artifacts in raw MEG data by detecting blinks, muscle artifacts, sensor jumps and cars passing by in the vicinity of the building. Blinks were detected based on concurrently recorded eye-movement signals (SR-Research EyeLink 1000). Sensor jumps were detected by convolving each sensor with a filter designed to detect large sudden jumps and subsequently by looking for outliers in the filter response. Muscle and environmental artefacts were detected by filtering each channel in the 100–140 Hz or <1 Hz range and by detecting outliers that occurred simultaneously in many channels. To remove power line noise, we applied a notch filter. In a final step, we epoched data, downsampled to 600 Hz and discarded all epochs that contained artifacts.

We computed time-frequency representations (TFRs) of single-trial data by using a multi-taper method. For low frequencies (1–9 Hz in steps of 1 Hz), we used a window length of 0.25 s (frequency smoothing of 8 Hz). For high frequencies (10–150 Hz in steps of 5 Hz), we used a window length of 0.1 s (20 Hz frequency smoothing).

We used linearly constrained minimum variance (LCMV) beamforming to estimate activity time courses at the level of cortical sources[60]. We constructed individual three-layer head models from subject specific MRI scans using fieldtrip[61] (functions, ft_volumesegment and ft_prepare_mesh). Head models were aligned to the MEG data by a transformation matrix that aligned the average fiducial coil position in the MEG data and the corresponding locations in each head model. Transformation matrices were generated using MNE software[62]. We computed one transformation matrix per recording session. Third, we reconstructed cortical surfaces from individual MRIs using Freesurfer and aligned two different atlases to each surface[63,64]. In a fourth step we used MNE[62] to compute LCMV filters for projecting data into source space. LCMV filters combined a forward model based on the head model and a source space constrained to the cortical sheet (4096 vertices per hemisphere, recursively subdivided octahedron) with a data covariance matrix estimated from the cleaned and epoched data. We computed one filter per vertex, based on the covariance matrix computed on the time-points from stimulus onset until 1.35 s after stimulus onset across all trials. We chose the source orientation with maximum output source power at each cortical location. In a final step, we computed TFRs of the epoched MEG data (same method as described for the sensor-level TFR decomposition) and projected the complex time-series into source space. In source space we computed TFR power at each vertex location and then averaged across all vertices within a ROI. We aligned the polarity of time-series at neighboring vertices, because the beamformer output potentially included arbitrary sign flips for different vertices.

For all analyses except the one for Supplementary Fig. 7b, we collapsed power values within hemisphere-specific regions of interest across vertices. We then converted the power values into units of percent power change (i.e., modulation) relative to the baseline power in each frequency[9]. To this end, we averaged power estimates across trials and time in a pre-stimulus interval (−250 to 0 ms before test stimulus onset). This yielded a frequency-specific, but condition-independent, baseline that was applied to normalize power values from each trial. The baseline was computed by averaging across all vertices within each hemisphere-specific part of a given region of interest (ROI; see below). For Supplementary Fig. 7b (fine-grained decoding), no pre-trial baseline was necessary due to feature standardization during the decoding analysis.

For all analyses except for Fig. 2c and Supplementary Fig. 7, the resulting baseline corrected hemisphere-specific power modulation values were collapsed or subtracted across hemispheres. The subtraction yielded the hemispheric lateralization of power modulations between the hemisphere contralateral and ipsilateral to the hand used to report stronger choices (choice-to-hand mapping counterbalanced across participants, see section *Task and Procedure* above).

**Regions of interest**. We used two sets of ROIs. The first contained 18 cortical regions listed in Table 1, all of which were defined by previous fMRI work: (i) retinotopically organized visual cortical field maps described in the atlas from Wang et al.[63] and (ii) three regions exhibiting hand movement-specific lateralization of cortical activity: aIPS, IPS/PCeS and the hand sub-region of M1[36]. Following a scheme proposed by Wandell and colleagues[65], we grouped retinotopic visual cortical regions with a shared foveal representation into clusters, thus increasing the spatial distance between ROI centers and minimizing the risk of signal leakage. The second, cortex-wide, set of ROIs were 180 regions covering the cerebral cortex, as defined in[64]. All ROIs were co-registered to individual structural MRIs.

**Table 1 Definition for the sensorimotor pathway.**

| Cluster | Functional areas | Source |
|---|---|---|
| V1 | Dorsal and ventral part of V1 | Ref. [63] |
| V2–V4 | Dorsal and ventral parts of V2, V3, V4 | |
| V3A/B | V3A, V3B | |
| IPS0/1 | IPS0, IPS1 | |
| IPS2/2 | IPS2, IPS3 | |
| Lateral occipital | LO1, LO2 | |
| MT+ | TO1, TO2 | |
| Ventral occipital | VO1, VO2 | |
| PHC | PHC1, PHC2 | |
| aIPS | aIPS1 | Ref. [36] |
| IPS/PostCeS | IPS/PostCeS | |
| M1 (hand) | M1 | |

**Choice-specific lateralization of neural activity**. We computed TFRs of choice-specific power lateralization for each ROI (contralateral vs. ipsilateral to hand movement used to report stronger choice). Lateralized activity was computed for each physical stimulus (i.e., mean test contrast) and choice condition separately before computing the final contrast that isolated choice-specific activity. For the latter, we contrasted error and correct responses for each physical stimulus (contrast) condition separately. We first averaged trials from each combination of stimulus and choice condition and subsequently computed the difference between hits and misses, and then the difference between false alarms and correct rejects, and finally averaged these two differences. In other words, we computed the difference between stronger and weaker choices, separately for each physical stimulus condition (mean contrast of test stimulus), and only averaged their results afterwards. This ensured that any activity differences due to the physical stimulus were factored out and the result isolated differential activity that was specific to behavioral choice. Statistical significance of lateralization values was assessed by cluster-based permutation test (threshold free cluster enhancement, $H = 2$, $E = 0.5$, $p < 0.05$).

**Contrast-dependent modulations of neural activity**. Stimulus-specific activity was computed similarly to the above choice-specific activity, but on TFRs of the hemisphere-averaged power values (not their lateralization). This was done because the full-field, centrally presented visual grating stimuli were expected to produce about equally strong sensory responses in both hemispheres. To factor out choice information in this analysis, conversely to the computation of choice-specific activity, we now contrasted different physical stimulus conditions (i.e., mean test contrast), separately for each choice and then averaged their result. Statistical significance of power changes was assessed by cluster-based permutation test (TFCE, $H = 2$, $E = 0.5$, $p < 0.05$).

**Decoding of choices**. We used multivariate pattern classification techniques[66] to decode choice information contained in the estimated activity patterns of individual ROIs. Decoding was carried out separately for each time-point throughout the trial in order to generate time courses of choice-predictive activity.

We used three different choice decoding approaches, referred to as approaches (i), (ii), and (iii) below. These approaches differed in their spatial extent and granularity, as well as in the treatment of the two physical stimulus categories (i.e., mean test contrast stronger than reference; mean test contrast weaker than reference). Approaches (i) and (ii) used the spectral pattern of power values (1–145 Hz) from both hemispheres for each ROI (i.e., coarse-grained spatial patterns), and we trained separate classifiers per stimulus category, effectively asking the classifiers to separate correct choices from errors within stimulus category. Thus, decoders from approaches (i) and (ii) could exploit hemisphere-and frequency-specific power values from each time-point and stimulus category to predict choices. In both approaches, features were z-scored before decoding based on training data only, and the same transformation was applied to test data before evaluating decoding performance. We used linear support vector machines for decoding ($C = 1$), as implemented in scikit-learn. Since choices were not equally distributed (<p_yes> = 0.57, <crit> = −0.23), we up sampled the minority class in the training data (but not in the test set) by randomly repeating elements until the frequency of choices was equal. In approach (i), the above-described procedure was applied to the visuo-motor pathway set of ROIs (Fig. 2c). In approach (ii), this procedure was applied to the cortex-wide set of ROIs (Supplementary Fig. 7a).

Approach (iii) used the spectral patterns of phase and power values from each individual vertex (fine-grained spatial patterns) for a subset of ROIs from the above two sets (Supplementary Fig. 7b). Owing to the high computational demand of approach (iii), we focused it on M1-hand (for comparison) and 17 pre-selected ROIs in dorsolateral prefrontal cortex anterior to premotor cortex, which have been implicated in different aspects of decision-making by previous work[67]. The procedure was as described above, except that we trained only one decoder to

predict choices across both stimulus categories, and that we performed dimensionality reduction after z-scoring. The latter was done due to the large number of features: two (power and phase) per vertex and frequency bin. We first computed principle components of the training set and kept all components that cumulatively explained 95% of the variance of the training set. We then projected training and test set data into the space defined by these components. We then evaluated a linear support vector machine with L1 penalty ($C = 10$/number of features) and discarded all features whose weights were below 1e-5. Finally, we trained another support vector machine (L2 penalty) on the final reduced feature set ($C = 1/2$) and plotted the performance of this classifier on the test set (Supplementary Fig. 7b). Principal component analysis and feature selection were performed exclusively on the training set and then applied to the test set.

In all three approaches, we evaluated decoding performance by means of 10-fold cross-validation and computed ROC-AUC values to evaluate each classifier and averaged across folds. We split all data per subject into ten folds, keeping the same percentage of choices in each fold. We then used nine folds to determine parameters of the classifier and computed prediction scores on the tenth fold, which we used to compute ROC-AUC values. We used Bayesian inference to estimate uncertainty around average decoding performance (error bars in Fig. 2b). We assumed that a participants AUC value at time-point $t$ were samples from a T-distribution (ignoring boundedness of AUC values because values were far from the bound) and placed weakly informative priors on all parameters of this distribution. We obtained posterior estimates using pymc3[68]:

$$\text{auc}_{s,t} = \text{StudentT}(\mu_t, \sigma_t, \gamma), \qquad (4)$$

$$\mu_t \sim N(0.5, 1), \qquad (5)$$

$$\sigma_t \sim U(0, 5), \qquad (6)$$

$$\gamma \sim \text{Exp}\left(\tfrac{1}{29}\right) + 1, \qquad (7)$$

where $s$ denoted the subject number and $t$ the time-point. StudentT denotes the T-distribution with mean $\mu_t$, standard deviation $\sigma_t$, and shape parameter $\gamma$. $N(0.5, 1)$ denotes a Gaussian distribution with mean 0.5 and standard deviation 1, $U$ the uniform distribution defined in the interval [0, 5], and Exp the exponential distribution. We used the NUTS sampler and two MCMC chains with 3000 iterations. We checked convergence visually and by ensuring that $\hat{R}$[69] was below 1.05.

We also compared decoding values in IPS/PostCeS after accounting for linear relationships with activity in M1-hand. We repeated decoding in IPS/PostCeS using single-trial power values across all frequencies. However, this time we first predicted IPS/PostCeS power values from their corresponding values in M1-hand using linear regression for each frequency separately. We subtracted this prediction from IPS/PostCeS power values and repeated the decoding procedure as before.

**Decoding of contrast or running mean of contrast**. We used a regression approach to assess whether source-reconstructed spectral activity tracked individual contrast samples or the running mean of contrast samples (termed accumulated contrast in Fig. 3). As the test stimulus was large and spanned both visual hemifields, we reasoned that decision-relevant neural activity in retinotopic areas should, likewise, span both hemisphere-specific parts of the visual field maps. Thus, individual contrast samples should be decodable from hemisphere-averaged activity in visual cortical field maps. As choice-predictive activity in downstream (pre-) motor regions was primarily contained in the hemispheric lateralization of power values contra- vs.- ipsilateral to the upcoming hand movement, we further assumed that the decision-relevant quantity, the running mean of contrast samples, should also be decodable from lateralized activity in choice-related areas. We, therefore, used either hemisphere-averaged power values from 1 to 145 Hz, lateralized power values (contralateral vs. ipsilateral to hand movement used to report stronger choice) for the results shown in Fig. 3.

In a separate analysis approach, we used separate power values from each hemisphere-specific part of each ROI for decoding of sample contrast or its running mean. This allowed the decoders to combine information from both hemispheres and use either the mean across hemispheres or the lateralization if beneficial. We found that this yielded almost identical results to those shown in Fig. 3, with very strong correlations of the decoding time courses (e.g., average correlations across subjects: 0.95 for decoding of sample contrast from V1; 0.91 for decoding of accumulated contrast from M1-hand). Decoding of accumulated contrast using lateralized power values yielded slightly better decoding performance ($t$-test at $t = 1.3$ s, peak for decoder that used both hemispheres separately: $p = 0.0003$, $t = 4.67$) presumably because the classifier had fewer but equally informative features at its disposal. Given that our choice to average or lateralize hemispheres was theoretically motivated from the outset and gave marginally better results, we decided to present the more constrained decoding results.

For each ROI, time-point $t$ and subject we extracted single-trial power values for all frequencies within the range from 1 to 145 Hz. We used hemisphere hemisphere-averaged or lateralized power at each time-point $t$ to predict the contrast of each sample $i$ ($i \in \{1..10\}$), or the running mean of contrast samples up to sample $i$, called accumulated contrast. This yielded one prediction per trial,

contrast sample position, time-point, subject, region of interest and hemisphere combination (averaged or lateralized).

We used ridge regression to predict target values (sample contrast or accumulated contrast). Power values were z-scored based on training data only and the same transformation was applied to test data before evaluating prediction performance. The analysis yielded the linear combination of frequency-specific power values that maximized the correlation between predictors and response variable (current sample contrast or accumulated contrast). The strength of L2 regularization applied by ridge regression was governed by a single parameter, which we optimized in a nested cross-validation (possible values were 0.1, 1, 10). We evaluated prediction performance by computing the Pearson correlation between predicted contrast and actual contrast. To decode the running mean of all contrast samples (accumulated contrast), we averaged the true contrast of all preceding and the current sample (i.e., up to sample $i$), assuming perfect evidence accumulation for simplicity, and predicted this value using the same power values. We used tenfold cross-validation in all cases (see above for more details).

**Regression of sample contrast on neural activity.** We also used linear regression to assess the sensitivity of activity at distinct frequencies to sample contrast. For this encoding analysis, we isolated single-trial power fluctuations at 190 ms after the onset of each contrast sample (identified as the peak latency of sample contrast decoding in the previous analysis), and for each frequency $f$ and sample position $i$ fit the following regression model:

$$p_{\text{trl},i,f} = \beta_1 c_{\text{trl},i} + \beta_0,\qquad(8)$$

where $p$ denoted power and $c$ sample contrast, both of which were normalized (z-scored) prior to fitting. The fitted coefficients $\beta_1$ reflected the strength and direction of the relationship between frequency-specific power and sample contrast (Fig. 3c). The reliability of time- and frequency-resolved contrast encoding at the group level was assessed via cluster-based permutation test.

**Computation of neural-activity kernels.** We computed so-called neural-activity kernels analogously to the psychophysical kernels in order to quantify choice-information contained in different frequency bands and ROIs. Kernels were computed by substituting single-trial sample contrast values with frequency and ROI-specific power values, again separately for both physical stimulus categories. Again, the resulting AUC values ranged between 0 and 1, with 0.5 indicating no association between neural activity and choice. See above section *Computation of Psychophysical Kernels* for the interpretation of AUC values larger or smaller than 0.5.

To maximize sensitivity for small-amplitude choice-predictive signals, that are expected in early visual cortex for such perceptual choice tasks (e.g., refs. [18,21]), we focused on the stimulus- and choice-specific frequency bands identified in previous analyses. Furthermore, we used hemisphere-averaged power values for neural-activity kernels because we assumed that decision-related activity in visual cortex would be distributed across both left and right parts of each visual field map. This assumption was motivated by (i) the stimulus spanning both visual hemifields and (ii) correspondingly, sample contrast decoding being nearly identical for hemisphere-averaged vs. separate-hemisphere signals (see previous section). Finally, we carried out a simple ROC-AUC analysis based on a single scalar variable, thus avoiding the need to fit free parameters for decoding this analysis. These choices made the analysis maximally similar to studies that computed choice probabilities of single neurons in primary visual cortex[18,21] and maximally sensitive for small-amplitude choice-predictive signals.

We used hemisphere-averaged gamma-band (collapsed across center frequencies 50–65 Hz, i.e., bandwith: 40–75 Hz; compare with Fig. 2b) or low-frequency band (center frequency: 10 Hz, bandwith: 0–20 Hz) power values extracted from visual cortical field maps for each sample position $i$ (at t = 190 ms after sample onset, i.e., average peak contrast encoding of individual samples) to compute ROC-AUC values. The so-computed kernels are referred to as overall kernels in Fig. 4.

A follow-up analysis aimed to further remove the effect of trial-to-trial fluctuations of the external contrast samples and isolate choice-predictive activity originating from intrinsic sources. Here, we regressed single-trial sample contrast values on the corresponding cortical power values (again at t = 190 ms after sample onset). We then subtracted this prediction from power values and used these residuals in the computation of V1 kernels, yielding the so-called residual kernels in Fig. 4.

For the cross-correlation with M1 choice decoding time courses (see next section), we also re-computed overall low-frequency kernels at the same (higher) temporal resolution (shown for V1 in Supplementary Fig. 10) as used for choice decoding in Fig. 2. To this end, we computed ROC-AUC values for each time-point $t$.

**Regression of alpha- and gamma-band activity on choice.** To complement the above neural-activity kernel analysis, we used logistic regression to model single-trial choice as a function of visual gamma-band (bandwidth 40–75 Hz) and low-frequency band (bandwidth 0–20 Hz) power and, importantly, their interaction. The interaction term in this model assessed the extent to which fluctuations in

low-frequency power altered the relationship between gamma-band power and choice. We first averaged power values at each frequency band (again at 190 ms post-sample onset) over the first (samples 1–5) and second (samples 6–10) half of contrast samples per trial, with the aim of mitigating noise in the single-sample responses but preserving our ability to assess temporal asymmetries between frequency bands. For each visual field map ROI, we then fit the following regression model:

$$y_{\text{trl}} = \text{logistic}\left[\sum_{h\in\{1,2\}} \beta_{1,h}\alpha_{\text{trl},h} + \beta_{2,h}\gamma_{\text{trl},h} + \beta_{3,h}\alpha_{\text{trl},h}\gamma_{\text{trl},h} + \beta_0\right],\qquad(9)$$

where $y_{\text{trl}}$ denoted the choice on trial trl, $\alpha_{\text{trl}}$ and $\gamma_{\text{trl}}$ denote the corresponding (z-scored) low-frequency and gamma-band activity, respectively, and $h$ denoted the half (first or second) of the test stimulus interval. We assessed the group-level significance of individual terms from this choice model via one-sample $t$-test against zero, computed temporal asymmetry scores by subtracting $\beta_{n,1}$ from $\beta_{n,2}$, and tested for differences in these temporal asymmetries between the two frequency bands via paired, two-sided $t$-test (Supplementary Fig. 9).

**Correlating choice-related activity in M1 and visual cortex.** We aimed to quantify the association between the time course of choice-predictive activity in M1-hand and low-frequency kernels in all visual cortical field maps. To this end, we computed, for each subject, the cross-correlation between the M1-hand choice decoding time course (ROC-AUC values from) and the high temporal resolution version of low-frequency kernels. Specifically, for each visual field map, we computed the Pearson correlation between the ROC-AUC values from M1-hand in the interval from $t = 0$ to $t = 0.8$ s and time-shifted versions (lags from −215 to 215 ms) of the low-frequency kernel for that visual field map (Fig. 5, inset). A lag of 100 ms, for example, correlated M1-hand ROC-AUC values in the interval $t =$ [0,0.8] $s$ with low-frequency kernels in the interval $t = [0.1, 0.9]$ s. This analysis yielded one correlation coefficient per lag, subject and visual field map. The cross-correlation functions for all visual field maps were finally collapsed, and tested for significance, across subjects (Fig. 5). Positive lags of the peak cross-correlation indicated a temporal advance of choice decoding performance in M1-hand. Negative lags of the peak cross-correlation indicated an advance of the low-frequency kernels in visual field maps.

Signal leakage between ROIs can confound correlations between estimates of regionally specific activity derived from source-reconstructed MEG-data[70]. Such leakage can occur due genuine field spread (volume conduction), as well as the limited precision of the LCMV spatial filters. To obtain an upper bound of the effect of signal leakage present in our data, we computed the zero-lag correlation between the ROC-AUC values of the M1-hand choice decoder and the low-frequency kernel, now taken from M1-hand, rather than the visual field maps. As this zero-lag correlation it compared two (partially redundant) signals obtained from the exact same ROI, it quantified the maximum possible leakage effects in our data. Thus, we used it as a conservative reference (Fig. 5, green) against which we compared the cross-correlation values obtained for the visual field maps (paired $t$-tests). Any significant difference pinpointed a component of visual field map cross-correlations that was unconfounded by signal leakage.

**Reporting summary.** Further information on research design is available in the Nature Research Reporting Summary linked to this article.

## Data availability

A reporting summary for this Article is available as a Supplementary Information file. The original data are provided as (i) source-reconstructed MEG data at [https://doi.org/10.6084/m9.figshare.12770366], (ii) as raw (sensor-level) MEG data at [https://doi.org/10.6084/m9.figshare.12759332], and (iii) as behavioral data only [https://doi.org/10.6084/m9.figshare.12783647]. Source data are provided with this paper.

## Code availability

The Python code for all analysis steps is available at https://github.com/DonnerLab/2020_Large-scale-Dynamics-of-Perceptual-Decision-Information-across-Human-Cortex.

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

## Acknowledgements
We thank Stanislas Dehaene, Mariano Sigman, and Genis Prat Ortega for discussion during this project, and Klaus Wimmer and Alan Stocker for thoughtful comments on the manuscript. This work has been funded by the Deutsche Forschungsgemeinschaft (DFG, German Research Foundation), grants DO 1240/3-1, DO 1240/4-1, and SFB 936 - Projekt-Nr. A7 (all to T.H.D).

## Author contributions
N.W.: conceptualization, methodology, investigation, software, formal analysis, visualization, writing—original draft, writing—review and editing; P.R.M.: conceptualization, methodology, writing—review and editing; F.M.: conceptualization, writing—review and editing; T.H.D.: conceptualization, methodology, resources, writing—original draft, writing—review and editing, supervision.

## Funding

## Competing interests
The authors declare no competing interests.
