## [Peer Review File · Nature Communications]

Reviewers' comments:

Reviewer #1 (Remarks to the Author):

This is an exciting and ambitious study with the potential to make a high impact and becoming of interest to a broad audience.

The experiments are novel and conducted to a high standard, and the analyses are appropriate.

My concern is in how some of the key results are presented and interpreted. I feel that the current way the results are presented and interpreted, though nice and easy to understand, may mislead readers and may turn off some of them, especially those coming from animal physiology or the computational ones. Fortunately, the problem is easy to address with a thorough, tighter and more conservative rewriting of parts of the manuscript. I think that this rewriting would pay off in terms of long-term impact of the study.

In particular, the main result of the authors is described (Abstract, Results, Discussion) as success in disentangling feedforward and feedback signals. The presentation of this as the main result in Abstract is of particular concern, as it may mislead readers.

Reading the paper, the actual result seems that high frequency activity and low frequency activity in specific regions have choice kernels with certain characteristics in both timing and strength after regressing out the contrast signal. The fact that some processes are more compatible with feedback processing and some other with feedforward is not a result. It is an interpretation. Turning this into a result would require a lot more work (for example, using statistical analyses such as Granger causality to show that low frequency activity is stronger in the feedback and in the feedforward direction, and so on). I feel that the results of the authors are strong and are interesting enough to be presented as they are, without being over-interpreted already in the abstract and results section.

I also have some comments about the decoding analyses. All analyses seem fair and good, but I am not sure that presenting results as slopes rather than decoding errors is the best decoding quantification. One problem is that this leads to have some arbitrary units in the y axis of the main decoding figures (e.g. Fig 3), and reporting results in a.u. should be avoided when possible. I wonder whether a decoding performance(%correct) would be better. Related to that, one would expect that the cumulative information about contrast accumulated over the different test stimuli periods should be at least as high as the instantaneous information extracted only from one test image period, but this seems not to be the case in V1 (Fig 3B). I am not saying that there is anything wrong in these analyses, but using more transparent analyses or addressing or better explaining some of the issues above would help readers like me.

Minor: Figure 2. In panel A, the caption states "left, stimulus locked responses, right choice locked responses". But this should be also made clear with some text inside the figure. I find it difficult to

understand what is plotted in each panel.

Page 10. The removing-effect-of-contrast analyses seem to rely on the assumptions that stimulus and choice signals are additive. This does not need to be the case, and the degree of additivity can vary between feedback and feedforward signals. Chicharro, Panzeri, Haefner have derived mathematically, and found in neural data, that choice and stimulus signal must interact in an inseparable and multiplicative way (both for feedback and feedforward processes, but possibly to a different degree depending on the details of the process) if the decision is taken by thresholding an internal evidence signal, as the authors assume in this study. It would be important that the authors make readers aware of the assumptions and limitations of their disentanglement analyses in Discussion, so that the readers can evaluate better how to interpret the author's results in terms of feedforward and feedback processes.

Reviewer #2 (Remarks to the Author):

Wilming et al. used MEG to identify spectral signatures of neural activity that are stimulus-related and choice-related during a perceptual decision-making task. They observed a gamma-band signal component (in visual areas) that ramps in proportion to stimulus contrast. They showed that later stages of processing (higher visual and decision/motor regions) are better correlated with an accumulated (rather than instantaneous) measure of the stimulus properties. They further observed that a ~10 Hz signal in early visual cortices provides ramping evidence of choice.

This paper was a great read. Overall, this experiment was well planned, the manuscript is very well written and argued, and the structure and presentation are clear. My critiques, below, focus on a set of methodological issues (related to band-limited and broadband effects, inter-trial interactions, and inter-regional couplings) that could undermine the main conclusions.

LARGER ISSUES

1) Please quantify whether gamma band power (GBP), alpha-band power (ABP) and high-frequency broadband power (HFBP) are ramping upward or downward over time within each region. This is a basic signal property which can affect the interpretation of the results. From Figure 2A, it would appear that GBP decreases in almost all visual channels, but it is not possible to be sure because of the color-scale ambiguities in PSDs, and this basic fact is never quantified. The change in amplitude in these signals-of-interest is important. For example: Figure 3 emphasizes that the decoding of contrast from V1 signals declines over time, but if the decoding of contrast in V1 is driven primarily by GBP, then surely the decrease in GBP (associated with a decrease in signal-to noise, SNR, for the decoder) is sufficient to explain the decrease in decoding? Adaptation effects are covered in the Discussion (and I agree these

are important to discuss) but for some reason this basic signal property (up/down ramping of GBP and ABP) is not, as far as I can tell, quantified in the Results.

2) The authors should summarize (and probably expand) the evidence that specifically supports an interpretation of inter-regional feedback processes, rather than a local recurrent process. For example, why could the ABP changes in V1 not arise from a process that is local to V1? Relatedly, it would appear that alpha power is present in V1 very early in each trial, even when it is not predictive of choice. So does that mean alpha is “performing different functions” in the early phase (where alpha is not choice predictive) and late phases (where alpha is choice predictive) in the trial? Perhaps it would be more parsimonious to assume that alpha processes reflected a local recurrent interaction [accumulation] occurring throughout the trial within V1 or a local visual circuit? It is important to summarize the evidence that specifically argues for inter-regional feedback, as opposed to some other kind of local feedback process.

3) Related to the previous point, the analysis of the M1-hand-to-V1 coupling (the final section of the Results) is critical and needs to be elaborated. This time-lag analysis provides the most direct evidence for feedback signaling; however, it is a very limited analysis, and should be elaborated to be more convincing.

First, the analysis is restricted only to V1-M1 pair — please test the same relationship between M1 and higher-order visual areas.

Second, basic information about the signal properties of M1-hand are not reported — please provide average spectral plots for M1-hand (averaged across all contrasts and choices) so that the time-course in alpha band and gamma band in M1-hand is clearly discernible; please report whether peaks are observed in M1 for the alpha/mu band and for the gamma band in the M1 baseline data.

Third, the time-lag analysis (Figure 4E) reveals that M1 is leading V1, but might this relate to differences in the detectability of signals supporting decoding in each area? In particular, if the M1 decoding is supported by (rapidly varying) gamma band signals and the V1 decoding is supported by (more slowly varying) alpha band activity — then might we not see a lag of V1 behind M1, simply because fluctuations in alpha power are detected (for signal processing reasons) later than fluctuations in gamma? This could potentially be dealt with by restricting the M1 decoding to also use the alpha band, and not the gamma band.

4) The authors indicate in the Discussion that they are aware of the debate around the theories of gamma-band signaling versus broadband power shifts. However, this Discussion seems not to address a critical issue: do the authors believe that their theoretical interpretation (feedforward signaling associated with band-limited gamma) would generalize to stimuli other than the drifting gratings used in this study? Many (most) real world stimuli do not elicit a sharp gamma response (most recently, see, e.g. Bartoli et al., 2019 and citation therein). So if the participants were performing an evidence accumulation task for a grayscale image which does not elicit a band-limited gamma response, would the authors expect to observe the same band-limited “feedforward” signaling in the gamma range?

Reference:

Bartoli, Eleonora, William Bosking, Ye Li, Michael S. Beauchamp, Daniel Yoshor, and Brett L. Foster. 2019. "Distinct Narrow and Broadband Gamma Responses in Human Visual Cortex." *BioRxiv*, 572313. <https://doi.org/10.1101/572313>.

5) Further related to the broadband/gamma distinction: can the authors rule out (or do they even wish to rule out) the possibility that the feedforward signals are also contained in broadband power? Could it be that feedforward signals are contained both in the gamma range, and also in higher frequencies, but the signal-to-noise ratio could be worse for higher frequencies, especially under standard MEG data processing schemes? It appears that the broadband response can be seen in MEG, but that it is much clearer following denoising:

Kupers, Eline R., Helena X. Wang, Kaoru Amano, Kendrick N. Kay, David J. Heeger, and Jonathan Winawer. 2018. "A Non-Invasive, Quantitative Study of Broadband Spectral Responses in Human Visual Cortex." *PLOS ONE* 13 (3): e0193107. <https://doi.org/10.1371/journal.pone.0193107>.

6) I was confused as to why hemisphere-averaged signals are used in the analysis of V1-choice kernels (Figure 4). In the methods, when choice-decoding is first described, the authors made clear that one might (in some cases) contra-lateral subtracted signals or (in other cases) multivariate signals available for decoding. Most of the manuscript (as far as I can tell) does not use hemisphere-averaged power for choice decoding. Why is hemisphere averaging used only in Figure 4? Do the results emerge in the same way when less constrained decoding procedures are used?

7) Please report whether choice on trial N can be decoded from any signals in the baseline period preceding trial N. This is important because it would provide a "negative control" for an alternative explanation of the ramping of alpha choice decoding (Figure 4). One could imagine that there is a slowly changing alpha power signal (changing on a scale even slower than a whole trial) and that this alpha signal reflects a drifting attentional bias. This drifting attentional bias could in turn give rise to differences in choice. This slowly-changing alpha power signal could be transiently masked at the time of stimulus onset (because of the wide-band energy elicited by signal onset) and which then re-emerges once the stimulus onset effect has cleared, and this unmasking shows up as the ramping alpha effect in Figure 4A (right panel). To rule this out, it would help to show that (under a different baselining scheme, of course) pre-trial alpha does not predict subsequent choice.

SMALLER POINTS

8) Please indicate on Figure 1 the actual times involved (e.g. that the 10 stimulus samples are presented over the course of 1 second); there is currently no sense of scale on the time-axis.

9) If 10 distinct stimuli are presented during the sampling period, are there onset/offset effects associated with the change in contrast between stimuli? Such onsets are not apparent at all in Figure 2, but would we not expect there to be some onset transients?

10) There is a relevant paper that should be cited and discussed: Pinto et al. (2019) recently presented an argument that evidence accumulation is not feedforward. The authors show that choice-related information is contained in widespread cortical regions, and account for their findings using a recurrent (rather than feedforward) neural network architecture. Please describe which aspects of this work you see as consistent / inconsistent with your MEG findings. Here is the reference:

Pinto, Lucas, Kanaka Rajan, Brian DePasquale, Stephan Y. Thiberge, David W. Tank, and Carlos D. Brody. 2019. "Task-Dependent Changes in the Large-Scale Dynamics and Necessity of Cortical Regions." *Neuron* 104 (4): 810-824.e9. <https://doi.org/10.1016/j.neuron.2019.08.025>.

Reviewer #3 (Remarks to the Author):

Authors study feedforward and feedback signals using MEG in a perceptual decision-making task. They report as a main result the representation of feedback signals at frequencies of about 10Hz in primary visual cortex. The study is based on a clever experimental paradigm, uses very sophisticated analysis tools and addresses a highly relevant research question. The manuscript is very well written and results are mostly presented in a convincing and clear manner. Still, I have some concerns regarding the presentation of results but mostly about the interpretation of the findings.

Major

- The main finding (as emphasised in title, abstract and discussion) is the identification of feedback signals in the alpha band in V1 (and maybe to a lesser degree the feedforward signalling in gamma band). However, I am not convinced that this is actually unambiguously shown in this study. First, most (if not all) studies presenting a dissociation of feedforward and feedback signalling into gamma versus alpha/beta frequency bands look at connectivity. Michalareas for example used granger causality. Here, authors infer feedforward/feedback signals purely from local activity in single areas (mostly V1). This demonstrates local coding of specific information (which is very nicely demonstrated) but does not directly demonstrate feedforward/feedback signalling. Second, Figure 4 is central for the claims made by the authors. Pertaining to the gamma-feedforward relationship we know that V1 gamma activity is related to stimulus contrast. The paradigm was designed in a way that stimulus contrast was relevant for task performance (and titrated to given above-chance performance level). So it is (nice to see but) not surprising that V1 gamma activity is related to choice. I had hoped to see effect on the residual gamma band kernel in Figure 4 which would be stronger evidence for a clear feedforward role. For V1 alpha activity, authors claim that it represents a feedback signal. Indeed, the time course resembles choice-

predictive activity in IPS and M1. Could the activity in V1 actually be a signal leakage from IPS (the beamformer can not perfectly separate different sources)? After all, similar activity is seen across the visual stream (Figure S4). It seems that unambiguous evidence of feedback would require directed connectivity or increasing feedback delays with increasing distance from higher order (feedback sending) areas. I am aware of Figure 4E but it is based on very coarse temporal sampling (100ms). Third, alpha and gamma effects are computed at 192ms after sample onset. This seems to be optimal for feedforward processing (peak time of contrast encoding) but is not necessarily optimal to study feedback signals. Authors might want to discuss this.

- Figure 2 is difficult to read. The separation of stimulus-locked and choice-locked data is almost invisible. I strongly suggest to change the graphics (making this separation more obvious) and explain in the caption clearly how this is represented (and the time window for the response-locked data (250ms?)).
- Can authors comment on the nature of the feedback signal? What would it signal to early visual areas? Is there any evidence in the data that strong feedback signals modulate the feedforward processing (e.g. on a trial-by-trial level)?

Minor:

- There are some typos or missing words in the manuscript that should be corrected.
- What does the shaded area in Figure 1B represent?
- I suggest to move at least some parts of Figure S1 to Figure 1. Behavioural data is important and it is a bit cumbersome to move back and forth between main text and Suppl Figs.
- I suggest to use box plots or violin plots for Figure S1A. Mean and SEM is not a good representation of individual data.
- In the main text it is difficult to understand the meaning of the word 'kernel' without reading the methods section. I suggest to explain it briefly when it is first used.

NCOMMS-20-05020: Response to reviewers

We thank the reviewers for their very constructive evaluation of our manuscript. We have addressed their concerns in a major revision effort. To this end, we have performed several new analyses, adding multiple new figure panels, one new main figure, and eight new supplementary figures. We have also substantially revised all sections of the manuscript, highlighted in **red font**. In what follows, we first provide a brief summary of our major revisions, followed by a detailed point-by-point reply to each of the reviewers' comments (printed in *blue italics*). We hope that you will now find our manuscript suitable for publication in *Nature Communications*.

Summary of major revisions

First, prompted by the comments of all Reviewers, we have revised large parts of Abstract, Introduction, Results, and Discussion, so as to more clearly separate the description of our results from their interpretation in terms of recurrent (within- and/or across-area) cortical processing. In this light, we also opted for changing the title to “Large-scale Dynamics of Perceptual Decision Information across Human Cortex”.

Second, prompted by Reviewer #2, we have also characterized in great detail the dynamics of several signal components per region (power in multiple frequency bands spanning the entire spectrum of the task-related response) across the trial. Among other things, this yielded even stronger correlations between physiology and psychophysical kernels, and hence a stronger case for the role of sensory adaptation in the decision process.

Third, we have expanded our analysis of the time-lagged correlations between downstream choice-encoding regions and early visual cortex, from a small single panel in former Figure 4 into a full new Figure 5 with corresponding supplement. Specifically, we applied the same analyses to visual cortical areas other than V1 to illuminate the cortical distribution of these coupling effects. We also devised new control analyses to rule out analysis confounds (e.g. signal leakage, processing differences for the local signals).

Fourth, we have realized that, although based on power estimates for 10 Hz, our reference to “alpha-band kernels” in the paper may have been slightly misleading. The reason is that the spectral smoothing used for this analysis was ± 10 Hz, so also including the delta, theta, and part of beta frequency ranges. This does not change our conceptual conclusions, especially because the kernel slope spectrum in Fig. 4D does show a clear peak in a narrow alpha-band. But because of this, we have opted to change our terminology to “low-frequency kernels” throughout.

Finally, in addition to these changes in response to the reviewers' comments, we have also added an analysis of choice decoding, separated by the two stimulus categories (stronger or weaker) *and* the confidence reports (high or low). This is now presented in Figure S6B, showing stronger choice-predictive activity, early during the evidence accumulation phase, for high compared to low confidence reports, further supporting our conclusion that the choice-predictive activity reflects accumulated evidence. Because we needed to restrict this analysis to 13 out of the 15 subjects (the remaining two had only a single trial in one of the eight conditions being compared, precluding decoding), we opted for presenting this result in the Supplement rather than the main paper. Nonetheless, we believe that it nicely complements the main analysis of choice-predictive activity in Figure S2.

We believe that our revisions have considerably strengthened the conclusions we can draw from our results, and that the sharper separation between the presentation of the results and our mechanistic interpretation improved the clarity of the paper.

Point-by-point reply to Reviewer #1

This is an exciting and ambitious study with the potential to make a high impact and becoming of interest to a broad audience. The experiments are novel and conducted to a high standard, and the analyses are appropriate.

We are delighted to hear you find the study exciting and ambitious.

My concern is in how some of the key results are presented and interpreted. I feel that the current way the results are presented and interpreted, though nice and easy to understand, may mislead readers and may turn off some of them, especially those coming from animal physiology or the computational ones. Fortunately, the problem is easy to address with a thorough, tighter and more conservative rewriting of parts of the manuscript. I think that this rewriting would pay off in terms of long-term impact of the study.

Thank you for raising this important issue. We fully agree with your assessment and have substantially revised the entire manuscript accordingly. Specifically, we now refrain from mapping of frequency-specific signal components onto feedback/feedforward signaling in the Results section of the paper, and have devoted a separate new paragraph in Discussion to unpacking this interpretation.

We would like to highlight that our analyses were guided from the outset by specific predictions for the qualitative features of choice-predictive activity in visual cortex, which were derived from a hierarchical circuit model of decision-making (Wimmer et al., 2015). We felt that it was important to keep making the theoretical rationale for each of our analysis steps in the final section of the results explicit. (Indeed, it was intriguing for us to find all the features of the feedforward and feedback components of choice-predictive activity in that model back in the kernels of our reconstructed V1 signal for low-frequency-/gamma-bands, respectively.) But, again, we agree that the mapping of frequency-specific signal components onto feedback/feedforward signaling remains an interpretation rather than result. So, we derived predictions and corresponding analyses from the feedback model, but avoid assigning the results of our analysis to feedforward/feedback interactions, respectively.

We hope that you will now find the separation between actual results and interpretation acceptable.

In particular, the main result of the authors is described (Abstract, Results, Discussion) as success in disentangling feedforward and feedback signals. The presentation of this as the main result in Abstract is of particular concern, as it may mislead readers.

Reading the paper, the actual result seems that high frequency activity and low frequency activity in specific regions have choice kernels with certain characteristics in both timing and strength after regressing out the contrast signal. The fact that some process is more compatible with feedback processing and some other with feedback is not a result. It is an interpretation. Turning this into a result would require a lot more work (for example, using statistical analyses such as granger causality to show that low frequency activity is stronger in the feedback and in the feedforward direction, and so on). I feel that the results of the authors are strong and are interesting enough to be presented as they are, without being over-interpreted already in the abstract and results section.

Again, we fully agree with the need to avoid over-interpretation, and have revised the text accordingly. We also agree with you, that the results presented in the current paper are strong as they are, but simply need to be presented more cautiously. Even so, we have now substantially expanded the lagged correlation analyses between the choice-specific (as well as non-specific) signal components

in different cortical regions. This provides a more exhaustive characterization of the correlation structure, including controls for signal leakage, which is now presented in Figures 5.

I also have some comments about the decoding analyses. All analyses seem fair and good, but I am not sure that presenting results as slopes rather than decoding errors is the best decoding quantification. One problem is that this leads to have some arbitrary units in the y axis of the main decoding figures (e.g. Fig 3), and reporting results in a.u. should be avoided when possible. I wonder whether a decoding performance(%correct) would be better.

We assume that this comment pertains to Figure 3 only: we use a fairly standard measure of decoding accuracy, area under the ROC-curve, in Figures 2 and 4. Please note that Figure 3 shows the decoding of a continuous variable (trial-to-trial variations in sample contrast). So, neither percentage of correct predictions nor area under the ROC-curve would work here: those apply to decoding of categorical variables (e.g. high vs. low mean contrast).

We opted to use the regression slope because we felt this measure is intuitively interpretable in the context of our study (reflecting change in decoded contrast per unit change in presented sample contrast). We now use the cross-validated (Pearson) correlation coefficient between predicted and presented contrast values, which is similarly interpretable. This measure of decoding precision provides qualitatively very similar temporal profiles to the originally used slope measure.

We would like to emphasize that the key issue for our study is not the magnitude of the decoding performance, but whether our analysis can detect, and discriminate between, the specific temporal profiles predicted for neural signals that track instantaneous vs. accumulated sensory evidence – our results show it can. This conclusion is supported by comparison of our results with those of conceptually analogous analyses of single-unit activity in monkey (Yates et al., 2017): Their Figure 2e matches our Figure 3 A, B quite closely.

Related to that, one would expect that the cumulative information about contrast accumulated over the different test stimuli periods should be at least as high as the instantaneous information extracted only from one test image period, but this seems not to be the case in V1 (Fig 3B). I am not claiming that there is anything wrong in these analyses, but using more transparent analyses or addressing or better explaining some of the issues above would help readers like me.

We apologize for the confusion. We realize that the description of our analysis in the corresponding part of the Results section was opaque, and that consequently there may have been a misunderstanding of the measures we decoded. We hope to have now resolved this by means of a more precise description, in both Results text and the Figure 3 caption (see also the following two paragraphs). To pre-empt confusion, we have now changed the short-hand labels from “*accumulated contrast decoding*” to “*decoding of accumulated contrast*” and from “*sample contrast decoding*” to “*decoding of sample contrast*” in the revised paper.

None of these measures pertains to decoding of behavioral choice; both pertain to decoding stimulus properties – either the contrast of individual samples, or the contrast averaged across samples. Specifically, *decoding of sample contrast* assesses how well a neural signal at time point t predicts the instantaneous contrast value that is presented at given sample position i (1-10, the different lines in Figure 3A). The contrast fluctuations across samples are conditionally independent. Thus, a neural signal that encodes the instantaneous input by means of a transient response should yield a transient boost of decoding precision, with some latency after each sample onset, and then quickly return back to baseline – precisely as we find for V1.

Conversely, *decoding of accumulated contrast* quantifies how well a neural signal at time point t predicts the running mean of the sample contrast presented up to and including sample i - i.e., the decision-relevant quantity in our task. (We call this *decoding of accumulated contrast* to highlight the analogy with metrics of “accumulated evidence” in other choice tasks). A neural signal that encodes the mean of contrast samples shown so far, should show more precise decoding of accumulated contrast than decoding of sample contrast, in line with what we find for IPS/PostCeS and M1. Conversely, a neural signal that only transiently responds to the contrast of individual samples without memory of previous samples should yield higher precision for *sample contrast decoding* than for *accumulated contrast decoding* – again, precisely what we find for V1.

Minor: Figure 2. In panel A, the caption states “left, stimulus locked responses, right choice locked responses”. But this should be also made clear with some text inside the figure. I find it difficult to understand what is plotted in each panel.

Agreed. We have now re-designed the figure to clarify what is shown.

Page 10. The removing-effect-of-contrast analyses seem to rely on the assumptions that stimulus and choice signals are additive. This does not need to be the case, and the degree of additivity can vary between feedback and feedforward signals. Chicharro, Panzeri, Haefner have derived mathematically, and found in neural data, that choice and stimulus signal must interact in an inseparable and multiplicative way (both for feedback and feedforward processes, but possibly to a different degree depending on the details of the process) if the decision is taken by thresholding an internal evidence signal, as the authors assume in this study. It would be important that the authors make readers aware of the assumptions and limitations of their disentanglement analyses in Discussion, so that the readers can evaluate better how to interpret the author’s results in terms of feedforward and feedback processes.

We agree. Another limitation, besides the one you mention, is that contrast response functions in the gamma-band (or other frequency bands) may not be perfectly linear in all individuals, although there is quite a bit of previous work indicating that this holds, at least on average (Hadjipapas et al., 2015; Henrie and Shapley, 2005). Indeed, we think the limitations of this regression analysis are potential reasons for the (apparent) lack of choice-predictive residual gamma-band activity in V1 (Figure 4). We have now acknowledged the limitations of the analysis in a new paragraph of the Discussion (p. 11, paragraph 3).

Thank you very much for your insightful and constructive comments.

Point-by-point reply to Reviewer #2

Wilming et al. used MEG to identify spectral signatures of neural activity that are stimulus-related and choice-related during a perceptual decision-making task. They observed a gamma-band signal component (in visual areas) that ramps in proportion to stimulus contrast. They showed that later stages of processing (higher visual and decision/motor regions) are better correlated with an accumulated (rather than instantaneous) measure of the stimulus properties. They further observed that a ~10 Hz signal in early visual cortices provides ramping evidence of choice.

This paper was a great read. Overall, this experiment was well planned, the manuscript is very well written and argued, and the structure and presentation are clear.

We are delighted to hear that you found our paper to be a great read.

My critiques, below, focus on a set of methodological issues (related to band-limited and broadband effects, inter-trial interactions, and inter-regional couplings) that could undermine the main conclusions.

1) Please quantify whether gamma band power (GBP), alpha-band power (ABP) and high-frequency broadband power (HFBP) are ramping upward or downward over time within each region. This is a basic signal property which can affect the interpretation of the results. From Figure 2A, it would appear that GBP decreases in almost all visual channels, but it is not possible to be sure because of the color-scale ambiguities in PSDs, and this basic fact is never quantified. The change in amplitude in these signals-of-interest is important. For example: Figure 3 emphasizes that the decoding of contrast from V1 signals declines over time, but if the decoding of contrast in V1 is driven primarily by GBP, then surely the decrease in GBP (associated with a decrease in signal-to noise, SNR, for the decoder) is sufficient to explain the decrease in decoding? Adaptation effects are covered in the Discussion (and I agree these are important to discuss) but for some reason this basic signal property (up/down ramping of GBP and ABP) is not, as far as I can tell, quantified in the Results.

Thank you very much for this perceptive comment. We now show the time courses of trial-related power changes in the theta (4-8 Hz), alpha (8-12 Hz), beta (12-45 Hz), gamma (45-65 Hz), and “high-frequency” (65-120 Hz) bands in a new Supplementary Figure S2. We show these time courses separately for the overall power (panel B) and the difference between stronger vs. weaker mean test contrast trials (panel C). For the overall responses, but not the differential response, there is indeed a clear down-ramping evident for both, the gamma- and high frequency-bands, yielding statistically significant, negative slopes of the gamma- and high frequency-band responses in V1 (right inset on top of V2-V4 in B). What is more, the gamma-band down-ramp is steeper when the sample-to-sample variance of contrasts a given stimulus sequence is smaller (Figure S3), linking these dynamics closer to sensory adaptation.

Intriguingly, we also find that the individual overall time course of power changes in gamma- and high-frequency-bands are strongly predictive of the individual psychophysical kernels, a result shown in the new Supplementary Figure S4. This was a surprising observation for us, because the overall power decay during the test stimulus presentation is a less specific signal (less directly related to the computation underlying the task) than the sample contrast decoding profile that we had originally focused on in our analysis of the link to evidence accumulation behavior (psychophysical kernels). Even so, the clear link to behavior that we find here indicates that the power decay is functionally meaningful. Combined with the dependence of the down-ramping of power on the sample variance, the correlation with evidence accumulation behavior provides evidence for the down-ramping reflecting some behaviorally significant form of sensory adaptation.

Results are different for the alpha-band. During the alpha-band power suppression that followed an initial positive transient (likely reflecting the phase-locked, evoked response), there is no monotonic change in power over the trial – if anything the power modulation is non-monotonic (Supplementary Figure 2, top right inset). There is a significant up-ramping of the post-transient suppression of beta-power (Supplementary Figure 2, top right inset).

Finally, as you suspected, the individual overall time course of V1 power changes in gamma- and high-frequency-bands (but not in other bands) are correlated with the V1 contrast decoding profile shown in Figure 3B (left, orange line), presented in Supplementary Figure S5. Indeed, this result is consistent with a reduction of SNR being a driver of the decline in decoding performance. For this reason, we have toned down the presentation of the decay in these decoding profiles, and focused our presentation of the decay on the underlying “raw signals”, i.e., the power time courses presented in Supplementary Figure 2. Please note that the correlation of decoding profiles with the power time courses does not, however, undermine the primary result of the decoding analysis: that we find significant contrast-specific V1 responses to each individual sample, from first to last, despite possible differences in SNR.

Again, we thank you for pointing us to this very important issue. We believe that our treatment of this point in the revised version has clarified and strengthened the paper.

2) The authors should summarize (and probably expand) the evidence that specifically supports an interpretation of inter-regional feedback processes, rather than a local recurrent process. For example, why could the ABP changes in V1 not arise from a process that is local to V1? Relatedly, it would appear that alpha power is present in V1 very early in each trial, even when it is not predictive of choice. So does that mean alpha is “performing different functions” in the early phase (where alpha is not choice predictive) and late phases (where alpha is choice predictive) in the trial? Perhaps it would be more parsimonious to assume that alpha processes reflected a local recurrent interaction [accumulation] occurring throughout the trial within V1 or a local visual circuit? It is important to summarize the evidence that specifically argues for inter-regional feedback, as opposed to some other kind of local feedback process.

Thank you for this comment. While the time-lagged correlations are consistent with inter-areal feedback, we clearly cannot (and do not want to) rule out a contribution from intra-areal (or “intra-system”) recurrence. We have now stated this possibility explicitly in the new Results section “Association between choice-predictive activity in visual cortex and downstream regions”. In fact, your point sits well with Reviewer #1’s request to tone down conclusions about feedforward vs feedback processing in general, which we have now done throughout the entire paper, from title and abstract to discussion.

3) Related to the previous point, the analysis of the M1hand-to-V1 coupling (the final section of the Results) is critical and needs to be elaborated. This time-lag analysis provides the most direct evidence for feedback signaling; however, it is a very limited analysis, and should be elaborated to be more convincing.

We have now substantially expanded this analysis, as described below.

First, the analysis is restricted only to V1-M1 pair — please test the same relationship between M1 and higher-order visual areas.

The cross-correlation plots are now presented for the full set of visual cortical field maps in the new Figure 5. We have also increased the temporal resolution of the analysis (including computation of low-frequency kernels) and added a schematic to the figure that illustrates the analysis procedure. Furthermore, we have added an explicit control for artefactual correlations due to spatial filter leakage in the source reconstruction (direct comparison of all cross-correlation functions with M1-M1 “reference” correlation, see main text).

Second, basic information about the signal properties of M1-hand are not reported — please provide average spectral plots for M1-hand (averaged across all contrasts and choices) so that the time-course in alpha band and gamma band in M1-hand is clearly discernible; please report whether peaks are observed in M1 for the alpha/mu band and for the gamma band in the M1 baseline data.

The average change in spectral power for M1-hand is shown in Figure 2A, rightmost panel. We now also show the corresponding baseline power spectra (for all areas, in the same format) in the new Supplementary Figure S2 (panel A). As expected, there is a peak in M1 for alpha/mu and a hint of a second peak for beta, but no evident peak across the gamma-band.

Third, the time-lag analysis (Figure 4E) reveals that M1 is leading V1, but might this relate to differences in the detectability of signals supporting decoding in each area? In particular, if the M1 decoding is supported by (rapidly varying) gamma band signals and the V1 decoding is supported by (more slowly varying) alpha band activity — then might we not see a lag of V1 behind M1, simply because fluctuations in alpha power are detected (for signal processing reasons) later than fluctuations in gamma? This could potentially be dealt with by restricting the M1 decoding to also use the alpha band, and not the gamma band.

This is an important issue, which we have considered carefully and addressed through simulations (Figure R1, also shown in the Supplement of the paper). Our simulations used a range of generic and noise-free scenarios (sustained, transient and ramping power increases, Figure R1, panels A-C, respectively) as well as more realistic ones containing noise (Figure R1, D/E). Based on the results, we consider this to be an unlikely explanation of the peak lags of 150 ms observed in the data for the cross-correlations between M1 choice decoding signal and low-frequency kernels in both, V1 and V2-V4 (Figure 5; we also find significant cross-correlations beyond 150 ms). Such time shifts are too big to be explained by conceivable signal processing confounds like the one you raise above.

First, choice decoding in M1 is largely supported by the lateralization of alpha- and beta-band power, not actually gamma-band power: Choice-related M1 gamma-band lateralization is detectable in stimulus-locked averages only just before the stimulus offset (Figure S6A), whereas changes in lateralized alpha- and beta-band power and decoding of accumulated contrast is detectable already ca. 0.5 s before stimulus offset (Figure S6A, Figure 2). Thus, differences in the detectability of power changes in different frequency bands for the two signals used as input for cross-correlation are unlikely to account for (artefactual) shifts of the true peak latency.

Second, even if there was a (weak) contribution from the gamma-band to the performance of the M1 choice decoder (and ignoring temporal smoothing due to spectral estimation, see next point), this could have only accounted for a maximum advance of M1 by 50 ms, relative to V1 (i.e., the low-frequency kernel). The full power increases in both bands are detected when the time-window for FFT evaluation covers a full cycle. This is first achieved when the FFT time window is centered on a latency

of half a cycle relative to signal onset. The difference between half the cycle durations for a 10 Hz signal (half cycle: 50 ms) and a 50 Hz signal (half cycle: 10 ms) like the ones we are considering here would be 40 ms (Figure R1 B,D). Even for a high-frequency signal of 100 Hz (half cycle: 5 ms), the difference would be smaller than 50 ms. Now, even this longest conceivable lag is far smaller than the observed peak lag of 150 ms.

Third, the above time shift would only occur for two correlated power transients (single cycle) in the two bands, regardless of the absence (Figure R1B, pink line in right plot) or presence (Figure R1D, pink line) of noise – such a time shift would neither occur for a sustained power increase in both two signals, nor for power ramps like the ones seen in our data (Figure R1, A, C, and E, pink lines on the right of each panel). In all these cases, the peak lags are indistinguishable from zero. This includes the most realistic scenario, a noisy signal with a ramping power increase (Figure R1, E).

Figure R1: Simulation signal-processing induced coupling delays. **A.** Schematic of simulation procedure. A signal oscillating at 10 Hz (1000 Hz sampling frequency; 2 s duration; “reference” in left panel) starts concurrently with a signal oscillating at 50 Hz (“Comparison” in left panel). Their respective power change over time is shown in the center left panel, evaluated in the same way as MEG power (multi-taper FFT; 100 ms time window for frequencies ≥ 10 Hz, 250 ms time for frequencies < 10 Hz; normalized to maximum of one). The cross-correlation between two signals yields a single peak (center right panel). Varying the frequency of the comparison signal shows how the peak-lag changes as a function of the frequency difference between both signals. We used a longer time window (400 ms) for frequencies below the 10 Hz reference signal, analogously to TFR parameters used for MEG data analysis. **B, C.** Same as A, but reference and comparison signals are comprised of only a single cycle. This yields positive lags for larger comparison frequencies and negative lags for smaller comparison frequencies. **D)** Same as C but with noisy signals (white gaussian noise: $\mu = 0, \sigma = 0.1$). **D, E.** Same as A, but using ramping signals. Panel E uses noisy signals (same as D). Error bars are ± 1 standard deviation of 250 simulation runs.

Fourth, when different window lengths are used for processing the two signals, as was partly the case in our analysis, the bias in estimated peak lag even shifts in the opposite direction (i.e., a lead of V1 low-frequency kernel relative to M1 choice decoding signal, Figure R1, see blue lines in all right

plots). In our analysis, spectral estimation parameters differed partially between the power time courses underlying both time courses used for cross-correlation: We used a time window of 100 ms for all frequencies from 10 Hz upwards and a time window of 250 ms for all frequencies below 10 Hz. So, we used a 100 ms window for calculating power time courses for the V1 low-frequency kernels (center frequency: 10 Hz), and also for calculating the power modulations in the alpha-/beta-, and/or gamma-bands feeding into the M1 choice-decoder. However, the M1 choice-decoder also included power modulations at center frequencies below 10 Hz, calculated with a time window of 250 ms. Our simulations show that this would, if anything, cause *later* detection of power changes by the M1 choice-decoder than the low-frequency kernel, hence predicting negative peak lags (V1 leads) in contrast to the positive lags (M1 leads) observed (Figure R1, blue lines).

4) The authors indicate in the Discussion that they are aware of the debate around the theories of gamma-band signaling versus broadband power shifts. However, this Discussion seems not to address a critical issue: do the authors believe that their theoretical interpretation (feedforward signaling associated with band-limited gamma) would generalize to stimuli other than the drifting gratings used in this study? Many (most) real world stimuli do not elicit a sharp gamma response (most recently, see, e.g. Bartoli et al., 2019 and citation therein). So if the participants were performing an evidence accumulation task for a grayscale image which does not elicit a band-limited gamma response, would the authors expect to observe the same band-limited “feedforward” signaling in the gamma range?

Reference:

Bartoli, Eleonora, William Bosking, Ye Li, Michael S. Beauchamp, Daniel Yoshor, and Brett L. Foster. 2019. “Distinct Narrow and Broadband Gamma Responses in Human Visual Cortex.” BioRxiv, 572313. <https://doi.org/10.1101/572313>.

We have now elaborated on this issue in the Discussion (p. 12, par. 2). We have also now analyzed the “high-frequency component” (65-120 Hz), likely reflecting the broadband activity measured in ECoG work, more carefully in the initial characterization of the physiological signals (Figures S2, 4, and 5). Also, we acknowledge explicitly in Discussion (p. 11, pars. 2 and 3) that frequency-specific power changes are likely only a proxy of the neural code used for the decision computation (the code being the pattern of spiking responses in the task-relevant neural populations).

Obviously, we can only make conclusive statements about the current grating stimuli used in the current study: it is possible that tasks with stimuli that do not induce robust gamma-band responses might also not show choice-predictive activity in the gamma-band.

That said, in many perceptual choice tasks, “raw” sensory signals are not directly fed into the decision computation, but first transformed into an internal representation of sensory evidence, which incorporates statistical knowledge (Gold and Shadlen, 2001). This evidence representation, the signal feeding into decision computation, may be associated with narrow-band gamma-band activity, regardless of the whether or not the underlying stimuli elicit sharp gamma-band responses. In another recent study using different visual stimuli (briefly flashed checkerboard patches, which do not elicit narrow-band gamma by themselves), we find that gamma-band responses across the visual cortical hierarchy encode sensory evidence (Murphy et al., bioRxiv 2020; currently under consideration at another journal). Clearly, more work is needed to illuminate this issue.

5) Further related to the broadband/gamma distinction: can the authors rule out (or do they even wish to rule out) the possibility that the feedforward signals are also contained in broadband power?

We certainly do not wish to rule out this possibility in general terms. But we have performed some analyses that shed light on this issue, for our particular data set.

First, we have performed a regression analysis, as a complement to the multivariate sample contrast decoding, to pinpoint which frequency bands most robustly encode the fluctuations in sample contrasts (Figure 3C). This shows clearest contrast encoding in the narrow gamma-band (45-65 Hz), but also statistically significant encoding in the high-frequency range (65-120 Hz). No other frequency band exhibits statistically significant contrast encoding.

Second, our frequency-dependent analyses indicate that at least the choice-predictive V1 activity is expressed in a narrow frequency band centered on 60 Hz, with little contribution beyond 80 Hz (Figure 4 C, D; “overall kernels”, blue).

In sum, in our current data, the feedforward signals seem to be contained in the higher frequency bands, and particularly in the narrow gamma-band. As discussed above, it is possible that the situation may be different for other types of stimuli.

Could it be that feedforward signals are contained both in the gamma range, and also in higher frequencies, but the signal-to-noise ratio could be worse for higher frequencies, especially under standard MEG data processing schemes? It appears that the broadband response can be seen in MEG, but that it is much clearer following denoising:

Kupers, Eline R., Helena X. Wang, Kaoru Amano, Kendrick N. Kay, David J. Heeger, and Jonathan Winawer. 2018. “A Non-Invasive, Quantitative Study of Broadband Spectral Responses in Human Visual Cortex.” PLOS ONE 13 (3): e0193107. <https://doi.org/10.1371/journal.pone.0193107>.

We are aware of the small amplitude of high-frequency activity in MEG, as well as the nice technical work the reviewer refers to. We now acknowledge the possibility of SNR limits in Discussion.

Yet, we would like to point out that we here used state-of-the-art signal analysis techniques. In particular, please note that the linear beamforming we applied here does not only yield ROI-specific activity estimates in source space, but the linear signal transformation also acts to suppress physiological noise, in particular muscle artefacts, which “live” in a subspace of the data that is largely orthogonal to the subspace spanning the ROI-level activity (Hipp and Siegel, 2013).

Indeed, in line with this notion, Figure 2A suggests that our current measurements reliably detect the broadband / high-frequency responses in visual cortex: there are no such signals in frontal regions, and the visual cortical responses overall are analogous to those from previous invasive ECoG recordings using similar (but smaller) drifting grating stimuli in monkeys (work from Pascal Fries’ lab) – if anything, the current responses show a stronger broadband component than observed in these monkey studies, which we attribute to the contrast fluctuations entailed in our stimuli.

6) I was confused as to why hemisphere-averaged signals are used in the analysis of V1-choice kernels (Figure 4). In the methods, when choice-decoding is first described, the authors made clear that one might (in some cases) contra-lateral subtracted signals or (in other cases) multivariate signals available for decoding. Most of the manuscript (as far as I can tell) does not use hemisphere-averaged power for choice decoding. Why is hemisphere averaging used only in Figure 4? Do the results emerge in the same way when less constrained decoding procedures are used?

Thank you for highlighting this. Your comment made us realize that our conceptual motivations for using hemisphere-averaged or hemisphere-lateralized power values, as well as our description of where these approaches were applied, were both opaque in the original version of the manuscript. In fact, we have focused on hemisphere-averaged and hemisphere-lateralized power modulations from the outset. We used to assess hemisphere-averaged power modulations to study the encoding of sensory input in retinotopic visual cortex because the stimulus was large and spanned both hemifields – thus, contrast dependent visual cortical responses are expected to occur across the retinotopic map.

We used hemisphere-lateralized power modulations to assess the encoding of the evolving decision in line with previous reports of the encoding of the plan for left vs. right hand movements in the hemispheric lateralization of activity across the downstream (pre-)motor cortical regions we have focused on here. The only places where we deviate from these constrained approaches were the “brute-force” choice decoding approaches shown in Figures 2C and S7. While the more constrained approaches were motivated from *a priori* considerations, we validated them for the current data through the application of less constrained decoding approaches reported in more detail now in Methods.

Specifically, we already used these constrained approaches for Figure 3. This shows data from a sample contrast decoder that uses hemisphere-averaged power values for V1 and lateralized power values for IPS/PostCeS and M1-hand. We also ran these analyses with decoders that utilized separate power modulation estimates from both hemispheres and found that the more constrained decoders performed either equally well, or slightly better. In particular, decoded contrast values were on average slightly larger when using hemisphere-averaged power values from V1 ($r_{avg} - r_{both} = 0.005$, $\sigma = 0.025$, and when comparing mean differences per sample they were so significantly (1-way repeated measures ANOVA with factor sample: $F=3.54$, $p=0.0006$). We have now stated this clearly in the Methods.

Based on the conceptual considerations, and based on the observation of sample contrast encoding in *hemisphere-averaged* V1 responses in Figure 3, we used the hemispheric averaging also for computing neural activity kernels in Figure 4. For this most critical analysis, the idea was to maximize (i) sensitivity for small-amplitude choice-predictive signals expected in visual cortex for such tasks (e.g. Britten et al., 1996; Nienborg and Cumming, 2009), as well as (ii) the correspondence with these single-unit physiology studies: a simple ROC-analysis based on a scalar variable (firing rate in their case, power in our case). The assumption here is the following: if, as implied by Figure 3 and the above-described control analyses, stimulus-encoding signals are distributed across both hemifields of the retinotopic maps (without any hemispheric asymmetry), then choice-related signals in visual cortex should be as broadly distributed. In addition, we assumed that choice-predictive signals would modulate only slowly in time, motivating the low temporal resolution used.

Using a less constrained, more highly-resolved, and cross-validated decoding procedure yields weaker choice-predictive activity in V1, presumably due to the small SNR combined with the broad spatial distribution of the underlying signals (Figure 2C).

7) Please report whether choice on trial N can be decoded from any signals in the baseline period preceding trial N. This is important because it would provide a “negative control” for an alternative explanation of the ramping of alpha choice decoding (Figure 4). One could imagine that there is a slowly changing alpha power signal (changing on a scale even slower than a whole trial) and that this alpha signal reflects a drifting attentional bias. This drifting attentional bias could in turn give rise to differences in choice. This slowly-changing alpha power signal could be transiently masked at the time of stimulus onset (because of the wide-band energy elicited by signal onset) and which then re-emerges once the stimulus onset effect has cleared, and this unmasking shows up as the ramping alpha effect in Figure 4A (right panel). To rule this out, it would help to show that (under a different baselining scheme, of course) pre-trial alpha does not predict subsequent choice.

Thank you for pointing us to this issue. We used the spectrum of the pre-stimulus activity, averaged across all trials (i.e. all combinations of stimulus category and choice), for baseline correction of the power responses analyzed in this paper. Because the same mean baseline values was applied to the power time courses from all trials, any slow signal modulation around that mean baseline values across trials that was related to choice should have been evident in the choice decoding time courses during the pre-stimulus-onset interval. By contrast, we do not find any significant choice-predictive activity

during the pre-stimulus baseline interval, in any cortical area. This is evident in the “unconstrained” decoding analysis shown in Figure 2C. We now also show the corresponding (more constrained and less conservative) ROC analysis for the baseline interval in Figure 4B, again showing no effect, neither in gamma- nor in the low frequency-band. In sum, we find no evidence for such a drifting choice bias in these neural data.

8) Please indicate on Figure 1 the actual times involved (e.g. that the 10 stimulus samples are presented over the course of 1 second); there is currently no sense of scale on the time-axis.

Done.

9) If 10 distinct stimuli are presented during the sampling period, are there onset/offset effects associated with the change in contrast between stimuli? Such onsets are not apparent at all in Figure 2, but would we not expect there to be some onset transients?

The gamma-band and high-frequency response time courses in Figures S2B and S3A do show periodic modulations of power consistent with sample onset responses. These transients are relatively small and therefore not apparent in the heat maps depicting the time-frequency profile of power responses in main Figure 2A.

10) There is a relevant paper that should be cited and discussed: Pinto et al. (2019) recently presented an argument that evidence accumulation is not feedforward. The authors show that choice-related information is contained in widespread cortical regions, and account for their findings using a recurrent (rather than feedforward) neural network architecture. Please describe which aspects of this work you see as consistent / inconsistent with your MEG findings. Here is the reference:

*Pinto, Lucas, Kanaka Rajan, Brian DePasquale, Stephan Y. Thiberge, David W. Tank, and Carlos D. Brody. 2019. “Task-Dependent Changes in the Large-Scale Dynamics and Necessity of Cortical Regions.” *Neuron* 104 (4): 810-824.e9. <https://doi.org/10.1016/j.neuron.2019.08.025>.*

Thank you for pointing us to this very relevant paper. We have now incorporated it into Introduction and Discussion.

Thank you very much for your insightful and constructive comments.

Point-by-point reply to Reviewer #3

Authors study feedforward and feedback signals using MEG in a perceptual decision-making task. They report as a main result the representation of feedback signals at frequencies of about 10Hz in primary visual cortex. The study is based on a clever experimental paradigm, uses very sophisticated analysis tools and addresses a highly relevant research question. The manuscript is very well written and results are mostly presented in a convincing and clear manner.

Thank you for the kind words on our study and the paper.

Still, I have some concerns regarding the presentation of results but mostly about the interpretation of the findings.

- The main finding (as emphasised in title, abstract and discussion) is the identification of feedback signals in the alpha band in V1 (and maybe to a lesser degree the feedforward signalling in gamma band). However, I am not convinced that this is actually unambiguously shown in this study. First, most (if not all) studies presenting a dissociation of feedforward and feedback signalling into gamma versus alpha/beta frequency bands look at connectivity. Michalareas for example used granger causality. Here, authors infer feedforward/feedback signals purely from local activity in single areas (mostly V1). This demonstrates local coding of specific information (which is very nicely demonstrated) but does not directly demonstrate feedforward/feedback signalling.

We agree with your assessment, and we have substantially revised the paper to address this issue. We have made substantial textual changes throughout the entire manuscript, so as to more clearly separate actual results from their interpretation. We also framed our interpretation in terms of cortical feedback interactions more cautiously, and have alluded to alternative possibilities. Finally, we have expanded the analysis of cross-correlations between choice-specific signals in different cortical regions, with some important controls, leading to a new Figure 5 and Results section (pp 9-10: “Association between choice-predictive activity in visual cortex and downstream regions”).

That said, we would also like to highlight that previous, highly influential studies in this field have likewise *inferred* interactions between cortical regions from the analysis of local activity (e.g. Gold and Shadlen, 2007; Hernández et al., 2010; Nienborg and Cumming, 2009; Siegel et al., 2015).

We understand that – like Reviewer #1 – you are not requesting that we actually perform explicit analyses of directed interactions, but we would like to explain the rationale for not going down that route in the current paper. Quantifying directed interactions between cortical regions based on the “Granger principle” will be a project in its own right, because it will require substantial new methodological work. Current analysis tools based on the Granger principle are not suitable for testing our hypothesis that choice-specific signal fluctuations in M1 predict choice-specific signal fluctuations in early visual cortex, for the following reasons. First, standard multivariate autoregressive modelling operates on the overall signal, of which only a small fraction is related to task-related experimental variables – in our case, choice. Second, it is not clear how to use the choice-specific signal time courses as input to multivariate autoregressive modelling, because these time courses (e.g. alpha kernels) are not defined at the single-trial level: they capture a property extracted by analyzing all trials from a given subject.

In contrast to standard Granger-like analyses, one novel analysis tool has specifically been developed for assessing “content-specific information transfer” (Bím et al., 2019). This opens up the exciting opportunity to assess these content-specific directed interactions in our data, but applying this approach will require a substantial number of choices and sanity checks, so that it is better reserved for a project in its own right. We hope that these considerations explain our hesitation to use “Granger causality analyses” for the current paper.

Second, Figure 4 is central for the claims made by the authors. Pertaining to the gamma-feedforward relationship we know that V1 gamma activity is related to stimulus contrast.

Indeed, previous studies have shown that V1 gamma-band activity tracks stimulus contrast, perfectly consistent with the result shown in our Figure 2C, and with the findings from many previous papers, which we discuss.

However, we would like to point out that the transient encoding of the rapidly fluctuating sample contrasts in V1 (Figure 3A/B, left) goes beyond the previous work: none of those previous studies of the contrast dependence of V1-gamma entailed contrast fluctuation as rapid as in our stimulus. It is not trivial that contrast at every sample position in such a rapid stream can be reliably tracked (e.g. on-/offset transients might “break” intrinsic rhythms).

To further flesh out this point, we have now complemented the decoding analysis with an encoding (i.e. regression) analysis in the new Figure 3C. This establishes that the gamma-band (and to a lesser degree: high-frequency band) contains the relevant contrast information.

The paradigm was designed in a way that stimulus contrast was relevant for task performance (and titrated to given above-chance performance level). So it is (nice to see but) not surprising that V1 gamma activity is related to choice.

We agree that this result may not be “surprising”, but please see our reply to the previous point: given the rapid nature of contrast fluctuations in our task, the result does not trivially follow from the previous work. The central relevance of our current finding lies in demonstrating this fine-grained, and rapid, encoding of contrast information, in the context of a precise quantification of the dynamics of evidence accumulation underlying behavior (psychophysical kernels), and most interestingly, the direct comparison between the neural dynamics associated with contrast encoding and the emerging decision.

I had hoped to see effect on the residual gamma band kernel in Figure 4 which would be stronger evidence for a clear feedforward role.

We agree. We have now elaborated on this issue in the revised Discussion (p. 11, par. 3). We believe that the lack of a choice-predictive effect in the residual gamma kernels should not be over-interpreted, due to the limitations inherent in the analysis we used to remove contrast fluctuations. This linear regression was based on the assumptions that (i) MEG power scales approximately linearly with contrast and (ii) stimulus-related and endogenous components superimpose linearly. Both assumptions are simplified: contrast response functions in MEG power may deviate substantially from linearity in some individuals (see Siegel et al. (2007) for a demonstration of motion coherence response functions), and stimulus- and choice-related cortical signals may interact multiplicatively (Chicharro et al., 2019). The absence of a choice-predictive component in residual gamma-band activity in V1 may be due to a failure of our linear approach to isolate the endogenous gamma-band fluctuations. It may also be due to the generally low signal-to-noise ratio of MEG gamma-band activity and/or due to gamma-band activity being only an indirect proxy of the neural code used for the decision computation.

That said, we do find weak choice-predictive activity, late in time during the trial, in extrastriate visual cortical areas V3A/B, IPS1/2, and LO1/2 (Figure S8). Given this subtle effect, we decided not to overemphasize this in the current paper. However, it is in line with the idea that spontaneous

fluctuations in gamma-band activity, later in the trial, and higher up in the visual cortical hierarchy shape the decision in a feedforward fashion.

For V1 alpha activity, authors claim that it represents a feedback signal. Indeed, the time course resembles choice-predictive activity in IPS and M1. Could the activity in V1 actually be a signal leakage from IPS (the beamformer cannot perfectly separate different sources)? After all, similar activity is seen across the visual stream (Figure S4).

Thank you for raising this important concern. We were confident that our result was not due to leakage because the M1 to visual cortex cross-correlation exhibited a clear peak at 150 ms (Figure 5). Leakage would predict a peak at zero lag, because leakage is instantaneous. We have now addressed this issue more directly with a new control analysis that aims to provide a conservative (upper bound) reference of the correlation component due to leakage present in the actual data. We referred to this as the “reference correlation” in Figure 5. The rationale behind the analysis was as follows. Whatever the exact nature of the leakage, its strength will decay monotonically with distance. We, thus, reasoned that the zero-lag correlation between the two signals *taken from the exact same area* (M1) should provide an upper bound for the effect of leakage on cross-correlation values measured between M1 and each of the visual cortical field maps in Figure 5.

Specifically, our cross-correlation analysis was based on two different types of signals: (i) choice-decoding time course for M1 (Figure 2C) and (ii) the low-frequency kernel for visual cortical field maps (Figure 4A). Thus, we used as a reference correlation the zero-lag correlation between the same signal types, substituting the V1 low-frequency kernels in (ii) by low-frequency kernels from M1. Direct comparison of the cross-correlations for visual cortical field maps with this M1-M1 (zero-lag) reference correlation showed significantly larger correlations for V2-V4 as well as several other downstream field maps, making it highly unlikely that leakage explains the observed effects.

Again, thank you very much for making us address this critical issue. We believe that our new analysis strengthens our conclusions substantially.

It seems that unambiguous evidence of feedback would require directed connectivity or increasing feedback delays with increasing distance from higher order (feedback sending) areas. I am aware of Figure 4E but it is based on very coarse temporal sampling (100ms).

We have now computed the low-frequency kernel at higher temporal resolution (Figure S10), enabling a more precise delineation of the peak lag in the cross-correlation functions shown in Figure 5. Indeed, we find stronger cross-correlations in higher-tier parietal regions (IPS2/3, aIPS) than for early visual cortex (compare Figure 5A and Figure S11A). We do not find any clear differences in the peak lags between these region pairs – this may be expected, given that the analysis is based on slower power modulations (Siegel et al., 2012).

We are agnostic about the specific “feedback sending” areas in our task, as well as about the pathways through which choice information in M1 is propagated to early visual cortex. The pathways may be quite different from those conveying the feedback of overall signal fluctuations across the visual cortical hierarchy exposed in the work by Bastos et al. (2015) and Michalareas et al. (2016). For example, the choice-predictive components of activity are clearly strongest in downstream, action-related regions including M1, which are not straightforwardly placed in the hierarchy of visual cortical regions.

Third, alpha and gamma effects are computed at 192ms after sample onset. This seems to be optimal for feedforward processing (peak time of contrast encoding) but is not necessarily optimal to study feedback signals. Authors might want to discuss this.

Since we have now increased the temporal resolution of the computation of low-frequency kernels for the cross-correlation analysis in Figure 5 (see previous reply), we circumvent the need to extract alpha-band power at any pre-selected latency (Figure S10).

We agree that the 192 ms latency is not well-motivated for low-frequency kernels. But we reasoned that it makes the low-frequency kernels directly comparable to the gamma-band kernels, for which this latency is well-motivated. This is why we had stuck to it for Figure 4.

- Figure 2 is difficult to read. The separation of stimulus-locked and choice-locked data is almost invisible. I strongly suggest to change the graphics (making this separation more obvious) and explain in the caption clearly how this is represented (and the time window for the response-locked data (250ms?)).

Thank you for pointing us to this. We have increased the gap between the two partitions of the plots.

- Can authors comment on the nature of the feedback signal? What would it signal to early visual areas? Is there any evidence in the data that strong feedback signals modulate the feedforward processing (e.g. on a trial-by-trial level)?

This is a very interesting and important question. We have attempted to touch on this issue through an additional analysis: We used logistic regression to quantify the impact of visual cortical alpha- and gamma-band activity on choice (Figure 4E; Figure S9). This revealed an analogous, qualitative pattern for the main effects of V1 alpha- and gamma-band activity as our ROC analyses – specifically, oppositely-signed temporal asymmetry of choice-predictive effects in both bands (Figure S9). Critically, the logistic regression enabled us to also model the interaction between these choice-predictive effects in both bands. This reveals a weak negative modulatory effect for the late interval, when low-frequency kernels are particularly strong, interestingly, specifically in V2-V4, not in V1 (Fig. 4E).

We are aware that our study is not suited (was not designed) to test for such interactions in the most sensitive and conclusive way. Nonetheless, we were intrigued to find this result, and thought it should be shared with the readership, hopefully to inspire future, in-depth analyses of these interactions.

- There are some typos or missing words in the manuscript that should be corrected.

Done.

- What does the shaded area in Figure 1B represent?

Thank you for pointing us to this – it is SEM across participants. This is now stated in the Figure caption.

- I suggest to move at least some parts of Figure S1 to Figure 1. Behavioural data is important and it is a bit cumbersome to move back and forth between main text and Suppl Figs.

We have moved panel D of that supplementary figure to main Figure 1. Our general rationale is to focus the characterization of behavior on the *dynamics* of evidence accumulation, leaving the quantification of the overall dependence of choice accuracy and confidence on contrast to the Supplement.

- I suggest to use box plots or violin plots for Figure S1A. Mean and SEM is not a good representation of individual data.

We now present individual data for that supplementary figure panel, in a similar format as used for the other figures depicting individual participants.

- In the main text it is difficult to understand the meaning of the word 'kernel' without reading the methods section. I suggest to explain it briefly when it is first used.

We have now clarified the meaning on first mention in Results.

Thank you very much for your insightful and constructive comments.

References

- Bastos, A.M., Vezoli, J., Bosman, C.A., Schoffelen, J.-M., Oostenveld, R., Dowdall, J.R., De Weerd, P., Kennedy, H., and Fries, P. (2015). Visual Areas Exert Feedforward and Feedback Influences through Distinct Frequency Channels. *Neuron* 85, 390–401.
- Bím, J., De Feo, V., Chicharro, D., Bieler, M., Hanganu-Opatz, I.L., Brovelli, A., and Panzeri, S. (2020). A Non-negative Measure Of Feature-Related Information Transfer Between Neural Signals. *bioRxiv*.
- Britten, K.H., Newsome, W.T., Shadlen, M.N., Celebrini, S., and Movshon, J.A. (1996). A relationship between behavioral choice and the visual responses of neurons in macaque MT. *Vis. Neurosci.* 13, 87–100.
- Chicharro, D., Panzeri, S., and Haefner, R.M. (2019). Stimulus dependent relationships between behavioral choice and sensory neural responses. *bioRxiv*.
- Gold, J.I., and Shadlen, M.N. (2001). Neural computations that underlie decisions about sensory stimuli. *Trends Cogn. Sci.* 5, 10–16.
- Gold, J.I., and Shadlen, M.N. (2007). The neural basis of decision making. *Annu. Rev. Neurosci.* 30, 535–74.
- Hadjipapas, A., Lowet, E., Roberts, M.J., Peter, A., and De Weerd, P. (2015). Parametric variation of gamma frequency and power with luminance contrast: A comparative study of human MEG and monkey LFP and spike responses. *NeuroImage* 112, 327–340.
- Henrie, J.A., and Shapley, R. (2005). LFP Power Spectra in V1 Cortex: The Graded Effect of Stimulus Contrast. *J. Neurophysiol.* 94, 479–490.
- Hernández, A., Nácher, V., Luna, R., Zainos, A., Lemus, L., Alvarez, M., Vázquez, Y., Camarillo, L., and Romo, R. (2010). Decoding a Perceptual Decision Process across Cortex. *Neuron* 66, 300–314.
- Hipp, J.F., and Siegel, M. (2013). Dissociating neuronal gamma-band activity from cranial and ocular muscle activity in EEG. *Front. Hum. Neurosci.* 7.
- Murphy, P.R., Wilming, N., Hernandez-Bocanegra, D.C., Ortega, G.P., and Donner, T.H. (2020). Normative Circuit Dynamics Across Human Cortex During Evidence Accumulation in Changing Environments. *bioRxiv*.
- Nienborg, H., and Cumming, B.G. (2009). Decision-related activity in sensory neurons reflects more than a neuron's causal effect. *Nature* 459, 89–92.
- Siegel, M., Donner, T.H., Oostenveld, R., Fries, P., and Engel, A.K. (2007). High-Frequency Activity in Human Visual Cortex Is Modulated by Visual Motion Strength. *Cereb. Cortex* 17, 732–741.
- Siegel, M., Donner, T.H., and Engel, A.K. (2012). Spectral fingerprints of large-scale neuronal interactions. *Nat. Rev. Neurosci.* 13, 121–34.
- Siegel, M., Buschman, T.J., and Miller, E.K. (2015). Cortical information flow during flexible sensorimotor decisions. *348*, 1352–1356.
- Wimmer, K., Compte, A., Roxin, A., Peixoto, D., Renart, A., and de la Rocha, J. (2015). Sensory integration dynamics in a hierarchical network explains choice probabilities in cortical area MT. *Nat. Commun.* 6, 6177.
- Yates, J.L., Park, I.M., Katz, L.N., Pillow, J.W., and Huk, A.C. (2017). Functional dissection of signal and noise in MT and LIP during decision-making. *Nat. Neurosci.* 20, 1285–1292.

****REVIEWERS' COMMENTS:**

Reviewer #1 (Remarks to the Author):

The authors did a great job in using the suggestions of all referees for further improving an already outstanding paper. I wish to congratulate the authors and I look forward to see the paper in press.

Reviewer #2 (Remarks to the Author):

I thank the authors for a detailed and thorough response, and commend them on an excellent manuscript.

I only have one minor (and entirely optional) comment remaining:

In light of the hint of $AUC < 0.5$ in Figure 4B, please add a note to the Methods indicating whether $AUC < 0.5$ would indicate pure noise (or mis-specified model) or whether it could indicate some kind of meaningful stimulus information. It is unclear to me whether such below-chance values must always (by definition, for this classifier scheme) reflect statistical noise.

Reviewer #3 (Remarks to the Author):

Authors have performed substantial new analysis and have addressed all my concerns. I am happy to recommend publication.

NCOMMS-20-05020A: Response to reviewer

We thank all reviewers for their very constructive evaluation of our manuscript, and for endorsing its publication.

Point-by-point reply to Reviewer #2

I thank the authors for a detailed and thorough response, and commend them on an excellent manuscript. I only have one minor (and entirely optional) comment remaining:

In light of the hint of $AUC < 0.5$ in Figure 4B, please add a note to the Methods indicating whether $AUC < 0.5$ would indicate pure noise (or mis-specified model) or whether it could indicate some kind of meaningful stimulus information. It is unclear to me whether such below-chance values must always (by definition, for this classifier scheme) reflect statistical noise.

We have now added the following sentences to the Methods section:

Subsection *Psychophysical kernels* (p. 11):

AUC values larger than 0.5 indicated that larger contrast values tended to be followed by “stronger” choices (with $AUC=1$ indicating perfect separation of contrast values between choices), and AUC values smaller than 0.5 indicated that larger contrast values tended to be followed by “weaker” choices.”

Subsection *Neural-activity kernels* (p. 15):

Again, the resulting AUC values ranged between 0 and 1, with 0.5 indicating no association between neural activity and choice. See above section Psychophysical Kernels for the interpretation of AUC values larger or smaller than 0.5.

These additions clarify that only AUC values around 0.5 reflect statistical noise, whereas deviations from 0.5 *in any direction* indicate systematic and meaningful associations between stimulus contrast or neural activity on the one hand and choice on the other hand.